# Winter-spring warming in the North Atlantic during the last 2,000 years: Evidence from Southwest Iceland

Nora Richter[1, 2], James M. Russell[1], Johanna Garfinkel[1], Yongsong Huang[1]

[1] Department of Earth, Environmental and Planetary Sciences, Brown University, Providence, RI 02912, USA

[2] The Josephine Bay Paul Center for Comparative Molecular Biology and Evolution, Marine Biological Laboratory, Woods Hole, MA, 02543, USA

*Correspondence to: Nora Richter (nora.richter@nioz.nl)*

**Abstract.** Temperature reconstructions from the Northern Hemisphere (NH) generally indicate cooling over the Holocene which is often attributed to decreasing summer insolation. However, climate model simulations predict that rising atmospheric $CO_2$ concentrations and the collapse of the Laurentian ice sheet caused mean annual warming during this epoch. This contrast could reflect a seasonal bias in temperature proxies, and particularly a lack of proxies that record cold (late fall-early spring) season temperatures, or inaccuracies in climate model predictions of NH temperature. We reconstructed winter-spring temperatures during the Common Era (i.e. the last 2,000 years) using alkenones, lipids produced by Isochrysidales haptophyte algae that bloom during spring ice-off, preserved in sediments from Vestra Gíslholtsvatn (VGHV), southwest Iceland. Our record indicates that winter-spring temperatures warmed during the last 2,000 years, in contrast to most NH averages. Sensitivity tests with a lake energy balance model suggest that warmer winter and spring air temperatures result in earlier ice-off dates and warmer spring lake water temperatures, and therefore warming in our proxy record. Regional air temperatures are strongly influenced by sea surface temperatures during the winter and spring season. SSTs respond to both changes in ocean circulation and gradual changes in insolation. We also found distinct seasonal differences in centennial-scale, cold-season temperature variations in VGHV compared to existing records of summer and annual temperatures from Iceland. Multi-decadal to centennial-scale changes in winter-spring temperatures were strongly modulated by internal climate variability and changes in regional ocean circulation, which can result in winter and spring warming in Iceland even after a major negative radiative perturbation.

## 1 Introduction

Temperatures in the Northern Hemisphere (NH) are generally thought to have cooled over the past 2,000 years, culminating in the Little Ice Age (LIA, c. 1450-1850 CE) (Kaufman et al., 2009; PAGES 2K Consortium, 2013, 2019; McKay and Kaufman, 2014). However, the majority of NH temperature reconstructions are based on proxies that respond to climate change during the warm season and may not capture trends in annual or winter and spring temperatures (Liu et al., 2014; PAGES 2K Consortium, 2019). This limits our understanding of major atmospheric phenomena in the NH, such as the North Atlantic Oscillation (NAO) which dominates wintertime variability, as well as changes in ocean circulation and other phenomena driving variability in the extent of Arctic sea ice.

Many oceanic and atmospheric processes that influence surface climate in the Atlantic and the broader NH are centered in the high North Atlantic region, making it an important location to study changes during the winter and spring seasons (Hurrell, 1995; Yeager and Robson, 2017). Terrestrial paleoclimate records from Iceland, for instance, have the potential to resolve temperature changes during the winter and spring seasons as this region is sensitive to the NAO and sits near the southern limit of Arctic sea ice (Hurrell, 1995; Hanna et al., 2004, 2006). The high sedimentation rates in Icelandic lakes, along with well-known volcanic eruptions that can be used as age constraints on sediment successions, make this an ideal location and archive to test how winter and spring temperatures evolved over the past 2,000 years (Larsen and Eiríksson, 2008; Geirsdóttir et al., 2009, 2019; Larsen et al., 2011; Langdon et al., 2011; Holmes et al., 2016). However, existing terrestrial records of temperature from Iceland are limited due to their sensitivity to the warm season, low temporal resolution and length, or compounding effects on proxies from human land-use or precipitation over the past 2,000 years.

Here we present a reconstruction of winter-spring temperatures developed using well-dated lake sediments from southwest Iceland to assess seasonal temperature changes in the North Atlantic climate over the past 2,000 years. We take advantage of alkenone-production by Group I Isochrysidales (i.e. haptophyte algae) during the spring season to develop a record of winter-spring temperatures and investigate the forcings responsible for cold-season temperature changes using a lake energy balance model.

## 2 Methods

### 2.1 Study site and age model

Vestra Gíslholtsvatn (VGHV) is a small lake (1.57 km$^2$) located in southwest Iceland (61 m a.s.l., 63° 56′ N, 20° 31′ W; Fig. 1), about 25 km from the coast (Blair et al., 2015). Mean monthly temperatures range from -1.4 °C during the winter months (DJF) to 10.4 °C during the summer months (JJA) (station at Hella, 1958-2005 CE; Icelandic Meteorological Office). Cores

were collected in 2008 using a Bolivia piston coring system (Blair et al., 2015), and were sampled at the National Lacustrine
Core Facility (LacCore) at the University of Minnesota.

The VGHV cores were dated using previously identified tephra, including seven historical and four pre-historical tephra beds
(Blair et al., 2015 and references therein). The age model was developed using 'classical' age modeling (CLAM) with a
smoothed spline fit (Blaauw, 2010). The resulting age model has an uncertainty of 5 to 15 yrs from -50 to 1200 yrs BP and 18
to 83 yrs from 1201 to 2000 yrs BP (Fig. 2).

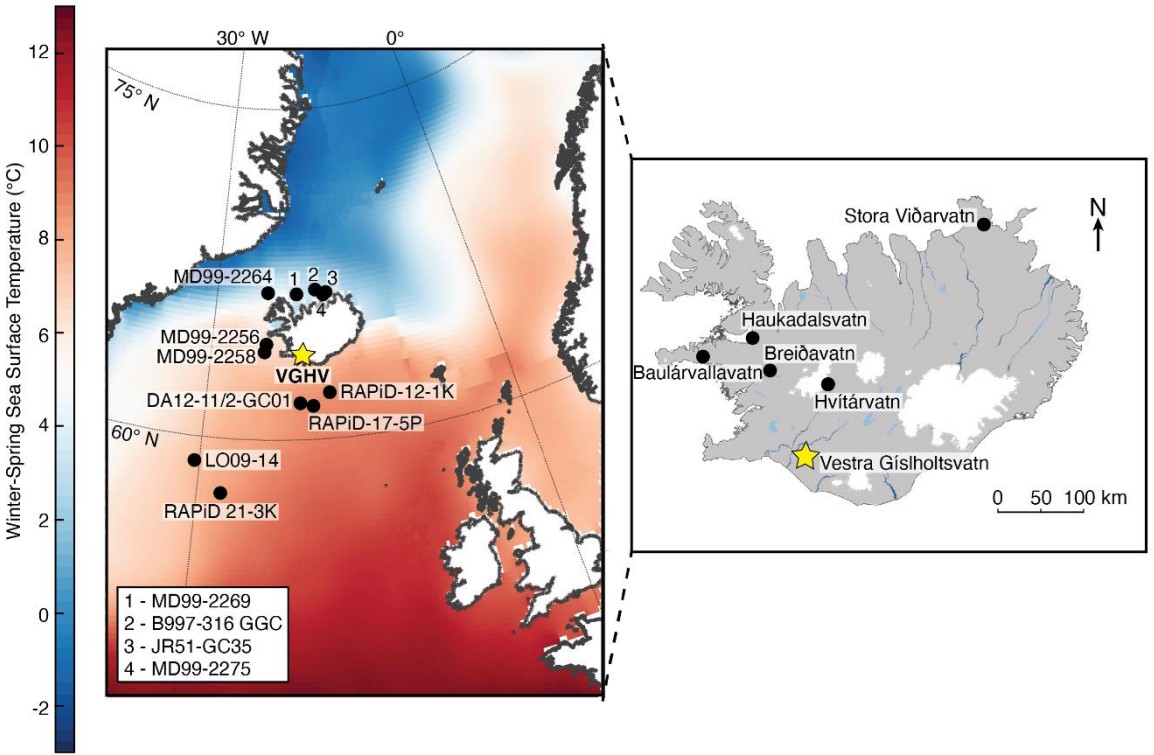

**Figure 1. Map of mean winter-spring (DJFMAM) sea surface temperatures from 1955-2017 in the high North Atlantic region. The marine sediment cores MD99-2269 (Moros et al., 2006; Justwan et al., 2008; Cabedo-Sanz et al., 2016), B997-316 GGC (Harning et al., 2019), JR51-GC35 (Cabedo-Sanz et al., 2016), MD99-2275 (Jiang et al., 2005, 2015; Massé et al., 2008; Sicre et al., 2008; Ran et**
**al., 2011), MD99-2264 (Ólafsdóttir et al., 2010), MD99-2256 (Ólafsdóttir et al., 2010), MD99-2258 (Axford et al., 2011), DA12-11/2-GC01 (Orme et al., 2018; Van Nieuwenhove et al., 2018), RAPiD-12-1K (Thornalley et al., 2009), RAPiD-17-5P (Moffa-Sánchez et al., 2014), LO09-14 (Berner et al., 2008), and RAPiD 21-3K (Sicre et al., 2011; Miettinen et al., 2012) that are discussed in the text are indicated. The locations of lake sediment records from Stora Viðarvatn (Axford et al., 2009), Haukadalsvatn (Geirsdóttir et al., 2009), Baulárvallavatn (Holmes et al., 2016), Breiðavatn (Gathorne-Hardy et al., 2009), and Hvítárvatn (Larsen et al., 2011) are**
**indicated. The study site, Vestra Gíslholtsvatn (VGHV), is marked by a yellow star. The maps were made using data from Natural Earth, the National Land Survey of Iceland, and the National Oceanic and Atmospheric Administration (NOAA) World Ocean Database (Boyer et al., 2018).**

## 2.2 Lipid analyses

Sediments were freeze-dried and extracted using a Dionex$^{TM}$ accelerated solvent extraction (ASE 350) system at 120 °C and 1200 psi. All of the extracts were separated by silica gel (40-63 µm, 60 Å) flash chromatography to obtain alkane (hexane; Hex), ketone (dichloromethane; DCM), and polar (methanol; MeOH) fractions. Saponification was used to remove wax esters by dissolving the dried ketone fraction in a 1 molar potassium hydroxide solution with MeOH:H$_2$O (95:5, v/v) and heating the samples for 3 hrs at 65 °C. 5 % NaCl in H$_2$O and 50 % HCl in H$_2$O were added to the samples and the lipid faction

was extracted using Hex (100 %). Ketone fractions were further purified using silver nitrate columns (D'Andrea et al., 2007) with DCM (100 %) followed by ethyl acetate (100 %) to elute the alkenones. If additional cleaning was needed, a modified procedure from Salacup et al. (2019) was used. The alkenone fraction was dried under N$_2$ gas and re-dissolved in 1.5 mL of DCM:Hex (2:1, v/v). To this, a 1.5 mL solution of 100 mg/mL urea in MeOH was added. The resulting crystals were dried under N$_2$ gas, and the urea addition was repeated two more times. The dried urea crystals were cleaned with Hex (100 %) and

extracted as the non-adduct. Milli-Q water was added to the vial to fully dissolve the urea crystals, and the adduct was extracted using Hex (100 %). The samples were then analyzed for alkenones. For several samples, co-eluting compounds were still present, or concentrations were too low for reliable quantification. These samples were not included in our final reconstruction.

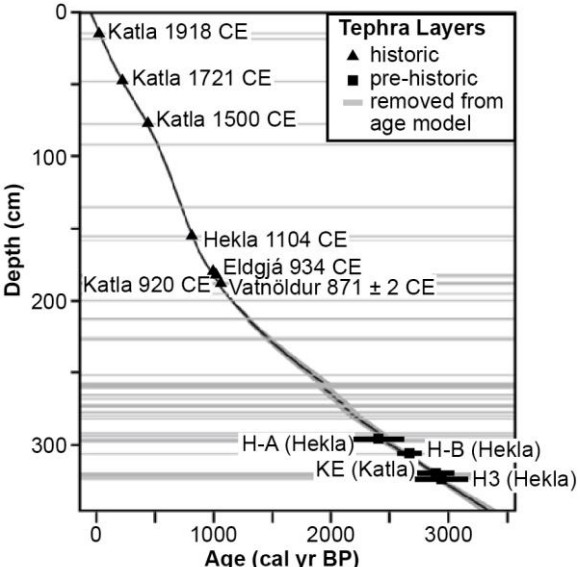

**Figure 2. Age model for Vestra Gíslholtsvatn with historic and pre-historic tephra layers (previously identified by Blair et al., 2015**
**and Christensen, 2013) indicated (figure adapted from Richter et al., 2021).**

The resulting alkenone fraction was analyzed using an Agilent 6890N gas chromatography (GC) and flame ionization detector (FID) system with an Agilent VF-200ms capillary column (60 m x 250 µm x 0.10 µm). Samples were injected into a CIS-PTV inlet in solvent vent mode (6.9 psi at 112 °C). The oven program was set to 50 °C and increased to 235 °C at 20 °C/min and ramped to 320 °C at 1.39 °C/min where it was held isothermally for 5 min. An alkenone standard was injected twice every 8 to 10 samples to assess the analytical precision of the alkenone measurements. The standard deviation for the calculated $U_{37}^K$ index (see equation 1) was 0.0040 (n = 28). For additional verification or identification of co-eluting compounds, samples were run on an Agilent 6890N GC system coupled with an Agilent 5793 N quadrupole mass spectrometer (MS). All samples were injected with pulsed splitless injection mode (20 psi at 315 °C) and run on an Agilent VF-200ms capillary column (60 m x 250 µm x 0.10 µm). The oven program was started at 40 °C for 1 min, ramped up to 255 °C at 20 °C/min, increased again to 315 °C at 2 °C/min, and then held isothermally for 10 min. The MS ionization energy was set to 70 eV with a scan range of 50 to 600 *m/z*.

## 2.3 Alkenones as a proxy for lake water temperatures

Alkenones are long-chain ketones produced by Isochrysidales haptophyte algae in both marine and lacustrine environments. Numerous marine-based culture and core-top studies show that variations in alkenone saturation (i.e., changes in $C_{37:3}$Me and $C_{37:2}$Me production) are inversely correlated with temperature, and can be linearly calibrated to temperature using either the $U_{37}^K$ or $U_{37}^{K'}$ index (Brassell et al., 1986; Prahl and Wakeham, 1987; Prahl et al., 1988; Müller et al., 1998; Conte et al., 2006). Similarly, culture studies, core tops, and in situ measurements in lakes show that changes in alkenone saturation are also correlated with temperature (Zink et al., 2001; Sun et al., 2007; Toney et al., 2010; D'Andrea et al., 2011, 2012, 2016; Wang and Liu, 2013; Nakamura et al., 2014; Longo et al., 2016, 2018; Zheng et al., 2016). The $U_{37}^K$ index can be applied to lacustrine environments and is calculated as follows:

$$U_{37}^K = \frac{[C_{37:2}Me] - [C_{37:4}Me]}{[C_{37:2}Me] + [C_{37:3}Me] + [C_{37:3*}Me] + [C_{37:4}Me]} \tag{1}$$

Sedimentary alkenones may derive from multiple alkenone-producing species, mainly Group I and II Isochrysidales, with distinct alkenone signatures and varying responses to temperature (Coolen et al., 2004; Sun et al., 2007; Theroux et al., 2010, 2013; Ono et al., 2012; Toney et al., 2012; Nakamura et al., 2014; D'Andrea et al., 2016). Group I Isochrysidales produces distinct tri-unsaturated alkenones (e.g., $C_{37:3*}$Me and $C_{38:3}$Et), which can be used to test for species-mixing effects with the $RIK_{37}$ and $RIK_{38E}$ indices (Longo et al., 2016):

$$RIK_{37} = \frac{[C_{37:3}Me]}{[C_{37:3}Me + C_{37:3*}Me]} \tag{2}$$

$$RIK_{38E} = \frac{[C_{38:3}Et]}{[C_{38:3}Et + C_{38:3*}Et]} \tag{3}$$

A RIK$_{37}$ value of 1.0 suggests a predominance of the C$_{37:3}$Me and the presence of Group II Isochrysidales, while values from 0.48 to 0.63 are empirically shown to correspond to Group I Isochrysidales (Longo et al., 2016, 2018). RIK$_{38E}$ values of 0 to 0.57 were empirically shown to correspond to Group I alkenone distributions, whereas Group II alkenone distributions correspond to values between 0.75 to 1 (Longo et al., 2016, 2018).

Group I Isochrysidales and their corresponding alkenones have, so far, only been identified in Northern Hemisphere lakes at latitudes ranging from 42-81 °N (Longo et al., 2018; Richter et al., 2019). The Northern Hemisphere lake calibration for Group I alkenones, which includes VGHV, was developed using the average spring temperatures for each lake during ice-off and the main Group I Isochrysidales bloom (U$_{37}^K$ = 0.029$T$-0.49, r$^2$ = 0.60, RMSE = ± 1.69°C; Longo et al., 2018). The calibration for Group I was updated to include additional lakes in northeastern China (U$_{37}^K$ = 0.030$T$-0.47, r$^2$ = 0.48) and now has an RMSE

= ± 1.71°C (Yao et al., 2019). Group I alkenone calibrations also exist for Lake BrayaSø in Greenland (U$_{37}^K$ = 0.025$T$-0.78, r$^2$ = 0.96, note the calibration also includes data from several German lakes, see Zink et al., 2001; D'Andrea et al., 2011), Lake Kongressvatnet in Svalbard (U$_{37}^K$ = 0.026$T$-0.80, r$^2$ = 0.85, D'Andrea et al., 2012), Toolik Lake in Alaska (U$_{37}^K$ = 0.021$T$-0.68, r$^2$ = 0.85; Longo et al., 2016), and Vikvatnet in Norway (U$_{37}^K$ = 0.028$T$-0.66, r$^2$ = 0.94; D'Andrea et al., 2016).

Temperature calibrations using the U$_{37}^K$ index were applied to develop high resolution records of summer temperatures in Greenland (c. 5,600 yrs BP; D'Andrea et al., 2011) and Svalbard (1,800 yrs BP; D'Andrea et al., 2012 and 12,000 yrs BP; van der Bilt et al., 2018, 2019), as well as a winter-spring temperature record in Alaska (16,000 yrs BP; Longo et al., 2020). The main timing of ice-off, and the corresponding alkenone bloom in VGHV, occurs in April to May in southern Iceland and May to June in northern Iceland (see Tables A2 & A3). In contrast, studies in Svalbard report ice-off dates between late June to

mid-August and show that alkenones primarily record summer (JJA) temperatures (D'Andrea et al., 2012; van der Bilt et al., 2019). In Northern Alaska ice-off dates primarily occur in June and reflect temperature changes during the winter and spring season (Longo et al., 2018, 2020). The regional variability in the relationship between the U$_{37}^K$ index and temperature, as well as differences in the timing of the alkenone bloom requires the development of local temperature calibrations and validation of the seasonal sensitivity of the proxy for each region (Wang and Liu, 2013; D'Andrea et al., 2016; Longo et al., 2016).

Unfortunately, there is currently no local calibration for Icelandic lakes, but in the following section we will describe how we use a lake model to test what drives changes in ice-off dates and lake water temperatures, and therefore the seasonality of temperature recorded by alkenones, in VGHV.

## 2.4 Seasonal temperature sensitivity of Group I alkenones in lakes

In Greenland and Alaska, Group I Isochrysidales bloom during the early spring in the photic zone as lake ice starts to melt
(D'Andrea and Huang, 2005; D'Andrea et al., 2011; Longo et al., 2016, 2018). Alkenone production starts prior to ice-off, then increases as the lake undergoes isothermal mixing, and decreases when thermal stratification begins to develop in late spring/early summer (Longo et al., 2018). This holds true for other Group I-containing lakes in the NH, including lakes in Iceland, as evidenced by the positive correlation between the $U^K_{37}$ index and mean spring air temperatures (Longo et al., 2018).

We investigated the controls on spring lake water temperatures and the timing of ice-melt in VGHV using a lake energy balance model (Dee et al., 2018). The purpose of the lake model was to determine the sensitivity of our proxy to different forcing mechanisms by assessing the temperature response and timing of ice-melt relative to our control simulation. Due to the lack of extensive observational datasets from VGHV to test our model, we adjusted the initial parameters using available data in the literature and parametrizations determined by Longo et al. (2020) for lakes in Northern Alaska where the lake model
was validated using limnological data from the Toolik Environmental Data Center and the Arctic Long-Term Ecological Research program over a 6-year period (see Table A1). Lake E5 in Northern Alaska is similar in size and with a similar catchment area to VGHV (Longo et al., 2020). The neutral drag coefficient was set to 0.002, and the albedos for slush and snow were set to 0.4 and 0.7, respectively. Note that volcanic eruptions in Iceland can result in ash deposits on the snow, and lower the albedo of the resulting slush and snow cover on VGHV and lead to earlier ice-off dates (Landl et al., 2003). However,
we expect this to only be important during volcanic eruptions that occurred during the winter and/or spring season and would only influence the lake water temperatures and ice-cover during that year. As our purpose is to understand how the lake responds on longer-timescales, we keep the values for albedo constant. The model was initialized using ERA-Interim daily data (1979-2018 CE; ECMWF; Dee et al., 2011) averaged over grid cells covering southwest Iceland (18.25° W-22.75° W by 63.00° N-64.50° N for a 0.75° x 0.75° grid). An initial control simulation was run for 39 years, followed by sensitivity tests
where various perturbations were introduced. Results from the control simulation were compared with available meteorological data, ice-off dates from nearby lakes in Iceland, and the few observations we were able to make using satellite imagery when our study site was not obscured by clouds (Tables A2 & A3; Fig. A2). The lack of extensive observational data from VGHV prevents us from validating the outputs from the lake model simulation, therefore we use the outputs from the lake model to highlight what processes could lead to variations in ice-off dates and lake water temperatures during the spring
season, but not to quantify the number of days or degrees that ice-off dates and temperatures, respectively, changed in the lake over the last 2,000 years.

The perturbation experiments focused on the effects of changes in seasonal air temperatures and shortwave and longwave radiation on lake surface temperatures and ice-off dates. We used instrumental data from Hella, Iceland (1958-2005 CE) to
determine the magnitude of seasonal air temperature changes (Icelandic Meteorological Office). Between 1958-2005 the range

of mean seasonal temperatures are as follows: winter (DJF) -3.7 °C to 1.8 °C, spring (MAM) -1.0 °C to 6.9 °C, summer (JJA) 8.8 °C to 12.0 °C, and fall (SON) -1.3 °C to 6.7 °C. To constrain the seasonality of our proxy, we perturbed the ERA interim seasonal air temperature values by -7 °C, -3 °C, 0 °C, +3 °C, and +7 °C and re-ran the lake model with the adjusted parameters. We repeated these experiments, but instead perturbed surface incident shortwave radiation to test how external forcings can drive changes in temperature. Incoming (top of the atmosphere) insolation at 63 °N has increased in winter (DJF) by 1.5 W m$^{-2}$ and in spring (MAM) by 3.7 W m$^{-2}$ over the past 2,000 years (Laskar et al., 2004). We therefore tested insolation forcing by perturbing seasonal changes in surface incident shortwave radiation by -4 W m$^{-2}$, -2 W m$^{-2}$, 0 W m$^{-2}$, +2 W m$^{-2}$, and +4 W m$^{-2}$. It should be noted that Iceland receives minimal light during the winter months and VGHV is frozen during the winter months, so we expect little to no direct influence of insolation on lake water temperatures during winter. Shortwave radiation values for the winter (DJF) were set to 0 W m$^{-2}$ if a negative perturbation decreased shortwave radiation below 0 W m$^{-2}$. To assess the effects of longwave radiation on lake water temperatures and ice-off dates, we decreased and increased incoming longwave radiation by -0.2 W m$^{-2}$, 0 W m$^{-2}$, +0.2 W m$^{-2}$. These values reflect the forcing from well-mixed greenhouse gas (GHG) radiation during the pre-industrial period (Schmidt et al., 2011).

## 3 Results

### 3.1 $U^K_{37}$ index: corrections for species-mixing

Our $U^K_{37}$ index from VGHV suggests that there was substantial variability in temperature during the last 2,000 years, but there was also variability in the community of alkenone-producers (Fig. 3a). Alkenones in VGHV surface sediments have a RIK$_{37}$ value of 0.60 and genetic analyses confirm that Group I Isochrysidales is the main alkenone-producer (Longo et al., 2018; Richter et al., 2019). However, the RIK$_{37}$ values increase slightly above the Group I cut-off of 0.63 about c. 500 CE, and then show a more sustained increase after human settlement in Iceland (c. 870 CE), suggesting that Group II alkenone-producers were also present in the lake (Fig. 3b).

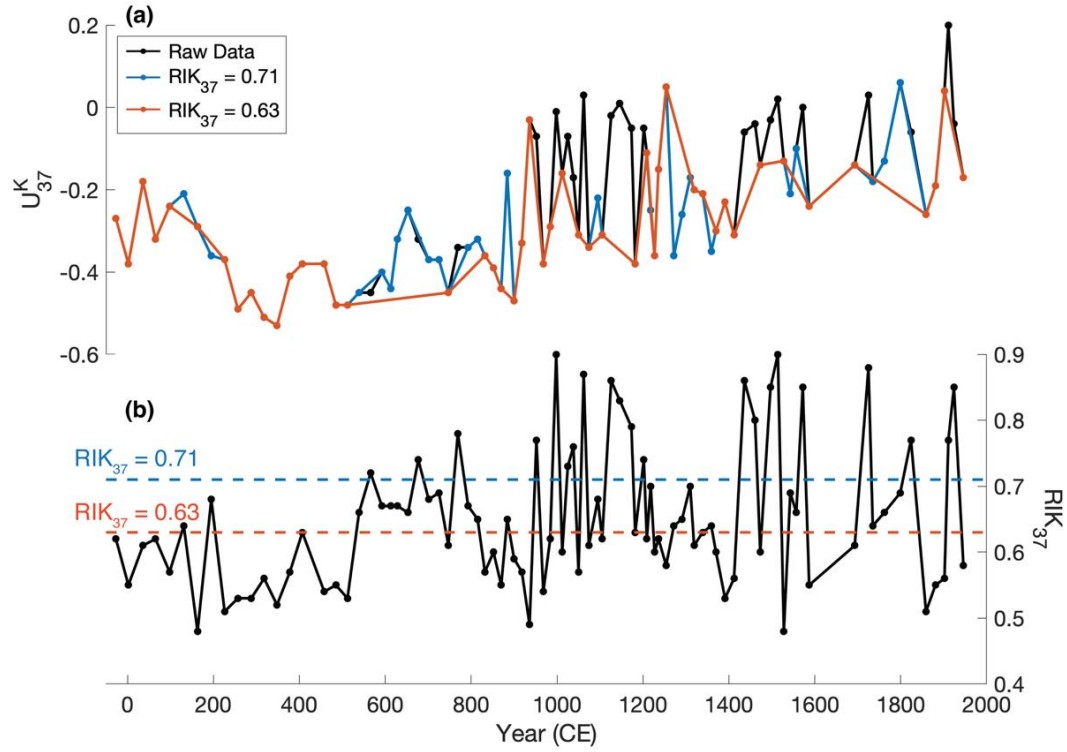

**Figure 3. (a) The raw data for the $U_{37}^K$ index is indicated in black, and the corrected $U_{37}^K$ index with a RIK$_{37}$ cut-off of 0.63 and 0.71 are shown in orange and blue, respectively. (b) The original RIK$_{37}$ index is shown below for comparison with the empirical cut-off, RIK$_{37}$ = 0.63, and cut-off for RIK$_{37}$ = 0.71 indicated.**

To evaluate the potential impacts of species mixing on the $U_{37}^K$ record, samples with a high abundance of Group II alkenones were removed. We tested several different cut-offs for the RIK$_{37}$ index and compared changes in the mean $U_{37}^K$ values (Fig. 4). If no correction is applied (RIK$_{37}$ = 1.0), then $U_{37}^K$ = -0.34 ± 0.12 from 0-1000 CE and $U_{37}^K$= -0.14 ± 0.13 from 1001-2000 CE. The empirically defined cut-off of 0.63 yields a mean $U_{37}^K$ index of -0.37 ± 0.12 from 0-1000 CE and -0.20 ± 0.10 from 1001-2000 CE. A less stringent RIK$_{37}$ cut-off at 0.71, results in no significant difference in the mean or the variability of the data (0-1000 CE $U_{37}^K$ = -0.36 ± 0.10 and 1001-2000 CE $U_{37}^K$ = -0.21 ± 0.11). Species mixing thus affects the $U_{37}^K$ temperature record, but regardless of the correction applied to the data, there is an increase in the mean $U_{37}^K$ values (which we interpret as warming) from 0-1000 CE to 1001-2000 CE. To further validate our results, we also did an additional comparison using both the RIK$_{37}$ and RIK$_{38E}$ indices (Fig. A1). Due to lower abundance of $C_{38:3}$ Et and its isomer we had fewer datapoints for comparison, but our results still demonstrate that datapoints used in our study are predominantly produced by Group I Isochrysidales.

Using a $RIK_{37}$ cut-off of 0.71, the corrected $U_{37}^K$ values and $RIK_{37}$ index are not correlated (r = 0.11, p = 0.35), indicating that
species-mixing effects do not affect the final temperature calibration. The resulting $U_{37}^K$ values can be interpreted as a record
of temperature changes from Group I alkenones.

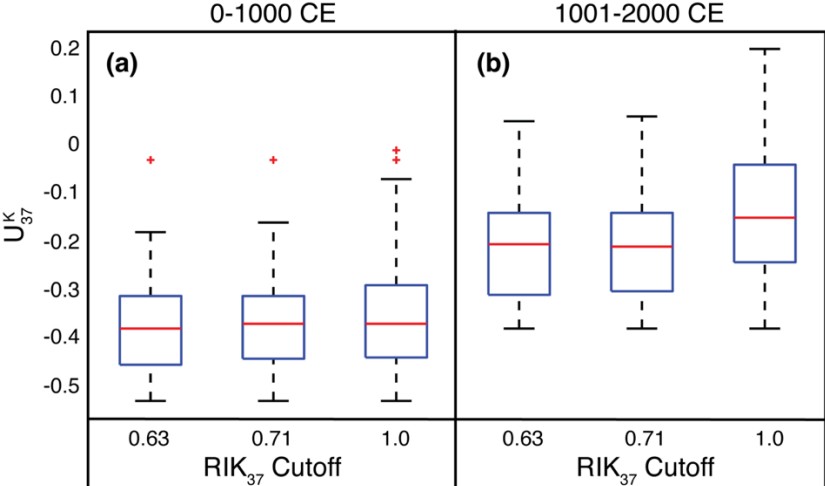

**Figure 4. Different $RIK_{37}$ cutoffs applied to the $U_{37}^K$ index for (a) 0-1000 CE and (b) 1001-2000 CE. A $RIK_{37}$ value of 1.0 indicates**
**that the data was not corrected for species-mixing effects, while $RIK_{37}$ = 0.63 corresponds to the empirically defined cut-off for**
**Group I and II (Longo et al., 2018).**

**3.2 Controls on spring lake water temperature**

Currently there are no extensive datasets on changes in lake water temperatures and/or ice-off dates for lakes (in particular
lakes that are not influenced by geothermal activity or are glacial lakes that are subject to sea water intrusions) in Iceland to
validate the control simulation from our lake model. However, a comparison with existing data in the literature and satellite
images that were not obscured by cloud-cover suggests that the timing of the ice-out dates in our lake model (mid- to late
April) and mean monthly temperatures of the surface lake water are reasonable (Tables A2 & A3; Fig. A2). We use the results
from the lake model to infer what processes could drive large changes in ice-off dates and lake water temperatures, and thereby
the determine the seasonal sensitivity of our proxy.

Results from the lake energy balance model show that seasonal perturbations can have a strong influence on spring lake water
temperatures and ice-off dates in VGHV (Fig. 5, Tables A4). The control run yields an average ice-off date of April 23rd with
surface water temperatures on May 1st of 4.1 ± 1.5 °C. Air temperature perturbations during the winter (DJF) and spring
(MAM) alter the timing of ice-off and how rapidly surface water temperatures warm, with warmer air temperatures leading to
warmer water temperatures and earlier ice-off dates (Fig. 5a-b). Changes in shortwave radiation did not have significant
influence on ice-off dates or lake water temperatures; the largest response is observed in spring (MAM) ice-off dates (Fig. 5c-

d, Table A5). Shorter days during the winter months (DJF) limits the amount of shortwave radiation reaching Iceland, and therefore has a minimal influence on Icelandic temperatures (Fig. 5d). There are no competing effects of summer (JJA) or fall (SON) insolation and air temperatures on spring lake water temperatures and the timing of ice-out. The increase in longwave radiation from GHGs during the pre-industrial period is relatively small and has no significant influence on either lake water temperatures (change from control = 0.0 ± 2.2 °C) or ice-off dates (change from control = 0.0 ± 8.1 days; Fig. A3, Table A6). Thus, the timing of the alkenone bloom and the water temperatures recorded by the alkenones are most likely responses to changes in air temperature during the late winter and spring season.

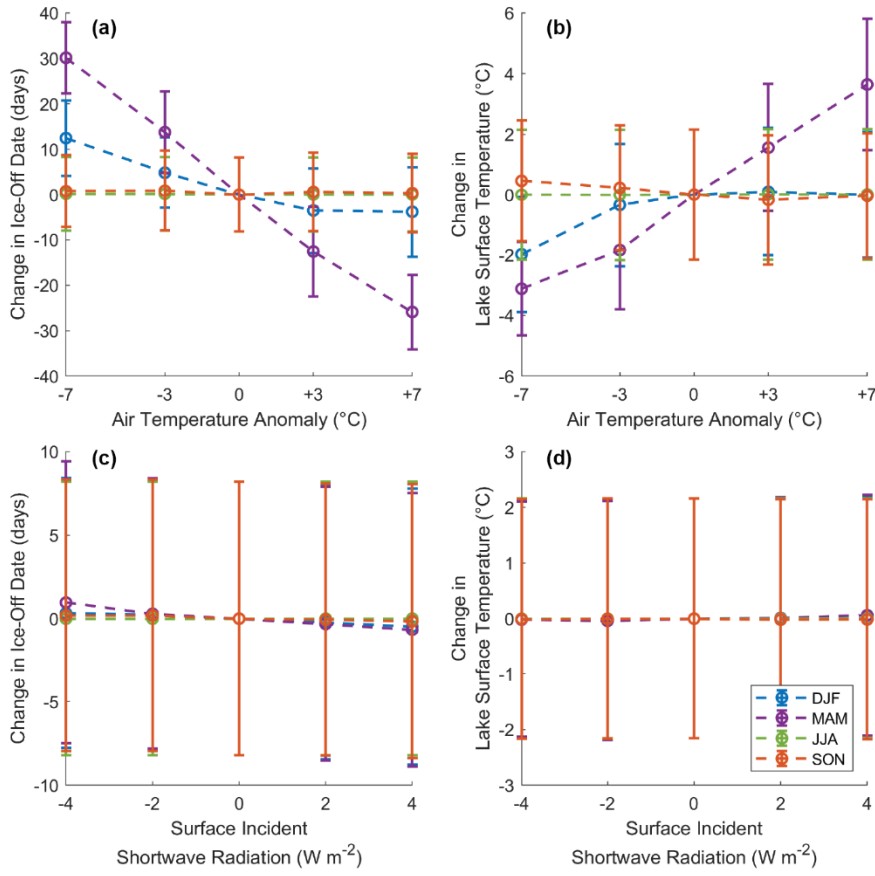

**Figure 5. Lake model sensitivity tests showing the impacts of air temperature perturbations during different seasons on (a) ice-off dates and (b) lake water temperatures on May 1st. Similarly, the affects of shortwave radiation on (c) ice-off dates and (d) lake water temperatures are shown. Changes in ice-off dates and lake surface temperatures were calculated relative to the control simulation. Seasonal changes are shown for winter (DJF, blue), spring (MAM, purple), summer (JJA, green), and fall (SON, orange).**

## 3.3 Long-term trends and short-term variability in the $U_{37}^K$ record and temperature

The $U_{37}^K$ index can provide temperature estimates using linear relationships that are calibrated in lakes with Group I alkenone-producers (D'Andrea et al. 2011, 2016; Longo et al., 2016, 2018). Existing temperature calibrations, except for the Northern Hemisphere calibration by Longo et al. (2018), for Group I are site-specific and therefore cannot be readily applied to VGHV (e.g. calibrations give estimates of 10.2 to 33.5 °C (D'Andrea et al., 2011), 7.1 to 34.4 °C (Longo et al., 2016), 4.4 to 24.5 °C (D'Andrea et al., 2016), -1.4 to 18.3 °C (calibration for Northern Hemisphere lakes by Longo et al., 2018; see Fig. A4). Most of the variation between sites is accounted for by the y-intercept of the calibration, so the slope of Group I calibrations was suggested as a better determinant of relative temperature changes for sites lacking a site-specific calibration (D'Andrea et al., 2016). However, the slopes determined for Group I calibrations still result in a very large and likely unreasonable temperature range of 26.9 °C for $U_{37}^K = 0.0219T$ (D'Andrea et al., 2016). The amplitude of the reconstructed temperature is still relatively large considering that each sample is an average of 5-19 years, and most likely stems from the lack of a local calibration. These discrepancies are highlighted by the variability observed in the Northern Hemisphere calibration developed by Longo et al. (2018), where only 60% of the $U_{37}^K$ index is explained by temperatures during the spring isotherm. In VGHV, the $U_{37}^K$-temperature relationship could highlight the differences in the sensitivity of VGHV lake water temperatures to air temperature relative to previous studies on Group I Isochrysidales (Longo et al., 2018). Despite these differences, the $U_{37}^K$ index is known to be highly sensitive to temperatures in NH lakes and both the millennial and multidecadal variability in our $U_{37}^K$ index exceeds the range of our analytical uncertainty. We assume that, after correcting for species mixing, there are no environmental parameters other than temperature that affect our $U_{37}^K$ record, and we therefore use the $U_{37}^K$ index to infer and evaluate qualitative changes in temperature trends and variability during the past 2,000 years.

The $U_{37}^K$ record from VGHV, corrected for species mixing, exhibits a long-term trend towards warmer spring lake water temperatures over the last 2,000 years as well as strong multi-decadal to centennial variability (Fig. 6). The gradual warming trend in our record begins after c. 400 CE. In particular, a warmer period occurs from the start of our record to c. 200 CE, followed by a cooler period from c. 250-600 CE. Temperature variability increases after c. 850 CE, and warmer periods occur between c. 850-1050 CE, c. 1100-1300 CE, and c. 1450-1550 CE. Relatively cooler periods occur at c. 1100-1200 CE, c. 1300-1450 CE, c. 1550-1750 CE, and c. 1850-1880 CE. However, caution should be used when interpreting results after c. 1400 CE because of low sampling resolution (c. 50 yrs between each sample).

## 4 Discussion

## 4.1 Regional controls on winter and spring temperatures in southwest Iceland

Results from the lake-energy balance model suggest that lake water temperatures during the spring season and the timing of ice-off in VGHV primarily respond to changes in air temperature during the winter and spring season. Air temperatures in southwest Iceland are typically warmer (Vestmannaeyjar mean annual air temperature (MAAT): 4.9 ± 0.6 °C for 1878-2002) than in northern Iceland (Grímsey MAAT: 2.3 ± 1.0 °C for 1878-2002) due to the advection of the warm Irminger Current along the southern coast and lack of sea-ice formation during the winter and spring seasons (Einarsson, 1984; Hanna et al., 2004).

2004). Based on our lake model simulation, spring surface water temperatures in VGHV respond to changes during the winter and spring season because the lake re-freezes every winter and reaches minimum lake water temperatures, meaning any influence from the previous summer or fall season are negligible (e.g. Assel and Robertson, 1995). Similar lake-model parameterizations were used to simulate spring lake water temperatures in a lake in Alaska, where the lake-model was rigorously tested against observational data, and the authors also observed that surface water temperatures during the spring

season only respond to air temperature changes during the winter and spring (DJFMAM) seasons (Longo et al., 2020). Further, an increase in winter and spring air temperatures leads to earlier ice-off dates. This is consistent with previous work in Alaska where an increase of 3 °C during the winter and spring (DJFMAM) led to an earlier simulated ice-off date by 12 days (Longo et al., 2020). Lake ice-off is expected to occur earlier during years when volcanic ash leads to "dirty" snow and lowers the albedo of the lake ice (Landl et al., 2003); however, this likely has a minimal influence on our record on 10 to 20-year

timescales. This suggests that our alkenone proxy is primarily sensitive to changes in air temperatures during the winter and spring season.

The VGHV temperature record shows that winter and spring air temperatures warmed over the last millennium, whereas temperature and sea ice reconstructions suggest that summer air temperatures cooled in Northern and Western Iceland as sea

ice increased and reached a maximum during the 18[th] and 19[th] centuries (Ogilvie and Jónsson, 2001; Moros et al., 2006; Massé et al., 2008; Gathorne-Hardy et al., 2009; Axford et al., 2009, 2011; Langdon et al., 2011; Cabedo-Sanz et al., 2016; Holmes et al., 2016). Paleo- and historical records, however, indicate that sea ice was only present along the southern and western coasts of Iceland, where our study site VGHV is located, during severe ice years when sea ice is advected clockwise around the country (Ogilvie, 1996; Axford et al., 2011; Cabedo-Sanz et al., 2016). Sea ice often leads to enhanced cooling in Northern

Iceland relative to Southern Iceland, leading to a larger temperature gradient and differences in the rate of warming, particularly during the winter and spring months (e.g. from 1871-2001 Grímsey in Northern Iceland warmed by 1.4°C and 0.6°C in winter (DJFM) and spring (MA), respectively compared to Vestmannaeyjar in Southern Iceland warmed by 2.1°C and 2.3°C in winter and spring, respectively; Hanna et al., 2004). In contrast to Northern Iceland, sea ice is therefore expected to have a smaller impact on air temperatures at our study site over the last 2,000 years.


Similar to our record from VGHV, millennial-scale changes in spring temperatures inferred from biogenic silica in western Iceland are decoupled from temperature and sea-ice changes in Northern Iceland, suggesting that spring temperatures are likely

more sensitive to changes in regional SSTs (Geirsdóttir et al., 2009). SST reconstructions near southern Iceland show that surface temperatures either increased (Berner et al., 2008; Thornalley et al., 2009; Miettinen et al., 2012; Orme et al., 2018) or

did not significantly change (Sicre et al., 2011; Van Nieuwenhove et al., 2018) over the last 2,000 years. Marine reconstructions of temperature from below the summer thermocline and bottom water record a decrease in mean annual temperatures over the Common Era (Thornalley et al., 2009; Ólafsdóttir et al., 2010; Moffa-Sánchez et al., 2014) as the transport of warm North Atlantic waters by the Irminger Current decreased over the last 2,000 years (Ólafsdóttir et al., 2010). The differences in proxy records could be related to differences in proxy seasonality. The different seasonal response in SSTs is observed in a gradual

increase in winter and decrease in summer temperatures over the Holocene in a marine record from southern Iceland, suggesting that SSTs near southern Iceland are sensitive to both changes in changes in ocean circulation and seasonal insolation (Van Nieuwenhove et al., 2018). As the maritime climate of southern Iceland is sensitive to changes in SSTs, we would therefore expect millennial-scale changes in both ocean circulation and insolation to be reflected in the VGHV temperature record. Based on existing paleo- and historical records we conclude that sea ice feedbacks play a minor role in driving long-

term changes in winter and spring temperatures at our study site, whereas an increase in SSTs along the southern coast could contribute to the warming trend observed in our record.

## 4.2 Long-term seasonal climate trends in North Hemisphere paleoclimate records

Mean annual temperature syntheses from the NH exhibit a long-term cooling trend over the last 2,000 years (Kaufman et al.,

2009; PAGES 2K Consortium, 2013, 2019) that is often interpreted as a response to decreasing summer insolation (Kaufman et al., 2009) and/or increased volcanic activity during the LIA (Miller et al., 2012). The magnitude of the cooling trend differs among global temperature reconstructions (PAGES 2K Consortium, 2019) and is often larger in NH temperature reconstructions compared to climate model simulations (Rehfeld et al., 2016; Ljungqvist et al., 2019). The discrepancies in temperature reconstructions and climate models could stem from a warm season bias in NH proxy reconstructions, leading to

an overestimation of changes in mean annual and cold season temperatures in proxy reconstructions compared to climate model simulations (Liu et al., 2014; Rehfeld et al., 2016; PAGES 2K Consortium, 2019).

The distinct warming trend in the winter-spring temperature reconstruction from VGHV could provide new insights into the mechanisms that influence cold season temperatures in the North Atlantic region. As discussed in the previous section, the

maritime climate of southern Iceland, and therefore the temperature recorded at our study site, reflect variability in seasonal SSTs that are driven by changes in ocean circulation and seasonal insolation. In contrast, mean annual long-wave radiative forcing, i.e. GHGs, over the pre-industrial period has a minimal influence on water temperatures and ice-off dates in VGHV. Long-term warming trends over the Holocene and the Common Era were also observed in other records of cold-season temperatures in the NH. For instance, pollen records of cold-season temperatures from North America and Europe (Mauri et

al., 2015; Marsicek et al., 2018) and an alkenone reconstruction of winter-spring temperatures from Alaska (Longo et al., 2020)

suggest that increasing winter and spring orbital insolation over the Holocene drove warming during the winter and spring season. Winter warming over the Holocene, including the Common Era, was also inferred from chrysophyte cysts in Spain (Pla and Catalan, 2005), ice-wedge records in the Siberian Arctic (Meyer et al., 2015; Opel et al., 2017), and a speleothem record from the Ural Mountains in Russia (Baker et al., 2017). A marine record directly south of Iceland from the North Atlantic subpolar gyre, records winter SSTs that are on average warmer over the last 2,000 years relative to the early to mid-Holocene (Van Nieuwenhove et al., 2018). In each of these studies, insolation and/or rising greenhouse gases are proposed as the primary mechanisms driving changes in seasonal temperatures.

Although there are existing reconstructions of NH winter and spring temperatures during the Common Era, the high-resolution reconstruction of winter-spring temperatures from VGHV is one of the few sites in the northern high latitudes where the effects of seasonal insolation on winter and spring temperatures can be tested without having to account for the influence of confounding factors, such as precipitation, on proxy records. For instance, varve thickness records have been interpreted to reflect winter temperatures but might be influenced by human activities in the catchment area, variations in snow accumulation, the timing of spring melt, and changes in precipitation (Ojala and Alenius, 2005; Haltia-Hovi et al., 2007). Varve thickness records from the Arctic that record changes in snow or glacial melt are also used to infer long-term cooling during the melt season; however, the melt season in the high Arctic often extends well into the summer months (Cook et al., 2009; Larsen et al., 2011) and can be affected by Arctic summer hydrology. Water isotope records from ice cores from Svalbard and a stalagmite from the Central Alps are sensitive to winter air temperatures during the instrumental period, but changes in the moisture source and seasonality of precipitation over time can alter long-term temperature interpretations (Isaksson et al., 2005; Mangini et al., 2005; Divine et al., 2011a, b). Although our record lacks a local calibration, we argue that the $U_{37}^K$ record from VGHV provides a robust, albeit qualitative, record of winter-spring temperatures given the unique seasonal growth ecology of Group I haptophytes in NH lakes. The warming trend observed in our record from Iceland and other records of cold-season temperatures from the NH suggest that long-term temperature changes are driven by a common forcing, orbitally-driven changes in winter and spring insolation, during the Holocene (Mauri et al., 2015; Meyer et al., 2015; Baker et al., 2017; Marsicek et al., 2018; Van Nieuwenhove et al., 2018; Longo et al., 2020) and during the Common Era (Pla and Catalan, 2005; Meyer et al., 2015; Baker et al., 2017; Opel et al., 2017).

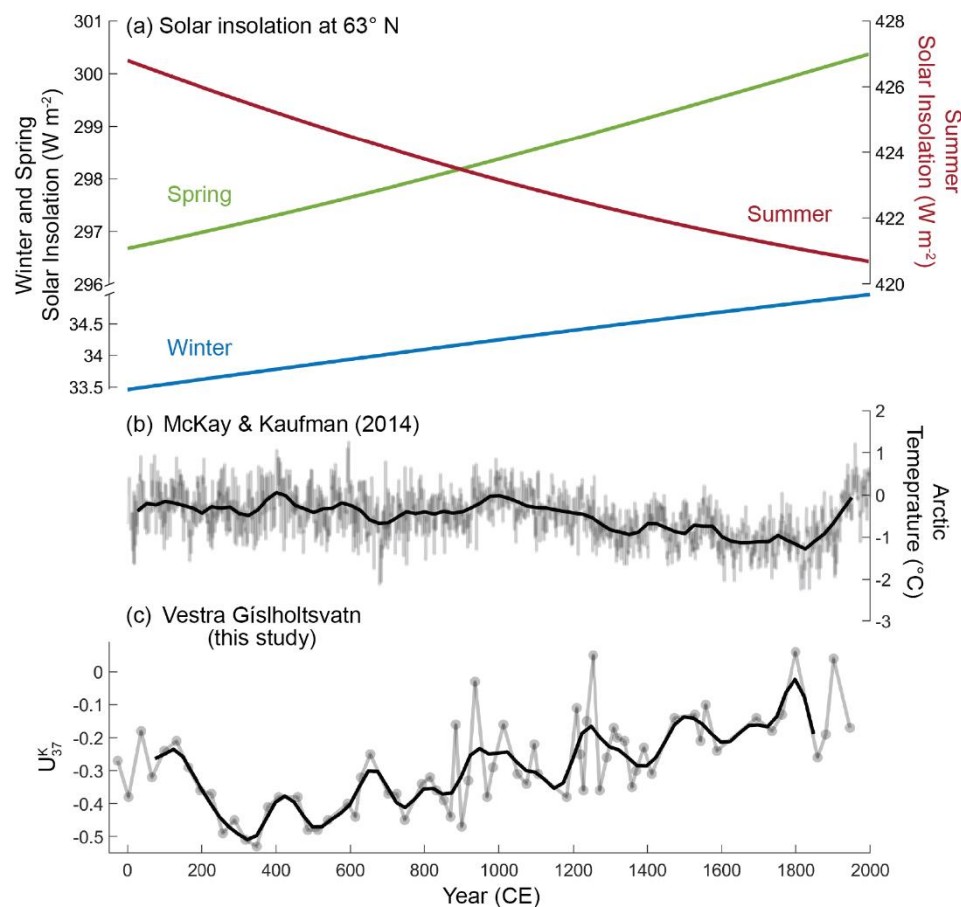

**Figure 6. (a) Changes in solar insolation at 63°N for winter (DJF), spring (MAM), and summer (JJA) are shown for the past 2,000 years (Laskar et al., 2004). In addition, (b) a compilation of Arctic temperature reconstructions (McKay and Kaufman, 2014) is shown for comparison with the (c) winter-spring temperature reconstruction from VGHV. The thick black lines correspond to lowpass filters (every 25 years) applied to the Arctic dataset and VGHV record.**

## 4.3 Multi-decadal to centennial variability in winter-spring temperature in the North Atlantic

The VGHV reconstruction of winter-spring temperatures provides an opportunity to test how changes in internal climate variability and external forcings (volcanic eruptions and total solar irradiance, or TSI) influence temperature changes during the winter and spring seasons in the North Atlantic region on multi-decadal to centennial timescales. The NAO, defined by differences in sea-level pressure between the subpolar low and the subtropical high, is a major source of atmospheric variability during the winter months (Hurrell, 1995). Although the NAO is mainly associated with interannual timescales, there is also

evidence that NAO-like patterns can emerge at multi-annual to centennial timescales, potentially linked to coupled changes in oceanic and atmospheric circulation (Visbeck et al., 2003; Delworth et al., 2016; Yeager and Robson, 2017). For instance, a positive (negative) NAO phase that persists for multiple winters can lead to increased (decreased) deepwater formation in the Labrador Sea and strengthening (weakening) of the subpolar gyre (SPG) and the meridional overturning circulation, thereby resulting in an increased (decreased) transport of warm waters towards the poles (Eden and Jung, 2001; Visbeck et al., 2003; Latif et al., 2006; Moffa-Sánchez and Hall, 2017). Alternatively, an increase in the southward transport of polar waters could also result in a reduction of deepwater formation in the Labrador Sea and a weaker SPG, leading to a decrease in northward oceanic heat transport and centennial cooling of ocean and regional air temperatures but warming of SSTs south of Iceland (Moffa-Sánchez and Hall, 2017; Moreno-Chamarro et al., 2017).

Whether forced or unforced, variability in winter atmospheric circulation, including the NAO, and sea ice extent are often linked to multi-decadal and centennial climate change in the North Atlantic region, particularly over Iceland (e.g. Hanna et al., 2006; Massé et al., 2008; Yeager and Robson, 2017). In the VGHV record of winter-spring temperatures there is a sharp decrease in the $U^K_{37}$ index c. 250 CE and a corresponding increase in drift ice along the North Icelandic shelf c. 400-900 CE (Fig. 7; Moros et al., 2006; Cabedo-Sanz et al., 2016). Cooling during this time period is typically attributed to volcanic eruptions and a minimum in solar activity (c. 400-700 CE; Fig. 7a-b); however, this cold interval was not uniform across the NH and records differ as to when peak cooling occurred (Helama et al., 2017). For instance, a colder interval is observed in a marine SST record from the south of Iceland c. 200-400 CE, whereas there is no distinct or prolonged cooling in SST records from the North Icelandic shelf (Fig. 7c; Sicre et al., 2008; Jiang et al., 2015) or in summer temperature records from Icelandic lakes (Gathorne-Hardy et al., 2009; Axford et al., 2009). In contrast, terrestrial records from the Arctic and Northern Europe indicate that temperatures were on average cooler between c. 450-700 CE and c. 500-650 CE, respectively (Kaufman et al., 2009; Sigl et al., 2015; Helama et al., 2017). The heterogenous temperature response in the North Atlantic region could be associated with a strengthening of the SPG between c. 200-400 CE, resulting in increased oceanic meridional heat transport to Northern Europe and colder SSTs near southern Iceland (Miettinen et al., 2012; Moffa-Sánchez and Hall, 2017). After c. 400 CE there is evidence of increased salinity and warming SSTs south of Iceland, that are associated with a gradual weakening of the SPG and contributed to a return to warmer temperatures in our VGHV record (Thornalley et al., 2009; Moffa-Sánchez and Hall, 2017; Moreno-Chamarro et al., 2017). A weaker SPG after c. 400 CE would also result in cooler Nordic waters and contribute to the cooler climate conditions observed in Northern Europe c. 500-650 CE (Miettinen et al., 2012; Helama et al., 2017; Moffa-Sánchez and Hall, 2017).

Winter and spring temperatures in VGHV were on average higher between c. 880-1100 CE than temperatures during the DACP but were not stable or particularly warm, unlike the climate usually associated with the Norse settlement of Iceland between c. 870-1100 CE (Fig. 7; Ogilvie et al., 2000). This time period corresponds to a peak in TSI and weak volcanic activity (PAGES 2K Consortium, 2013), and warmer summer and annual SSTs near Northern and Southern Iceland (Sicre et al., 2008, 2011;

Justwan et al., 2008; Ran et al., 2011). Summer temperature reconstructions from lakes in Northern and Western Iceland also record warmer temperatures c. 800-1300 CE but with distinct cold excursions occurring between c. 1000-1300 CE (Axford et

al., 2009; Gathorne-Hardy et al., 2009; Holmes et al., 2016). Peaks in sea ice, however, are only noted after c. 1200 CE (Ogilvie, 1992; Ogilvie and Jónsson, 2001; Massé et al., 2008; Cabedo-Sanz et al., 2016), suggesting that an alternative mechanism, such as the NAO, may be responsible for the short-term variability observed in terrestrial temperature records from Iceland during this time period.

The 14[th]-15[th] centuries mark the start of the LIA and are often associated with a colder-than-average climate in Iceland (Ogilvie, 1984; Ogilvie and Jónsson, 2001). In contrast, the VGHV record indicates there was no prolonged cold period during the winter and spring season at our study site between c. 1300-1800 CE, which could indicate a continued response to the warming observed in SSTs south of Iceland (Thornalley et al., 2009; Miettinen et al., 2012). Cooling during this time period is associated with major volcanic eruptions and decreases in TSI (PAGES 2K Consortium, 2013; Otto-Bliesner et al., 2016) that led to an

increase in sea ice (Fig. 7; Moros et al., 2006; Massé et al., 2008; Cabedo-Sanz et al., 2016) and cooling of summer/annual SSTs between c. 1400-1900 CE along the North Icelandic shelf (Fig. 7c and e; Jiang et al., 2005, 2015; Sicre et al., 2008; Ran et al., 2011). A record of subsurface winter temperatures inferred from glycerol dialkyl glycerol tetraethers (GDGTs) along the North Icelandic shelf also record cold excursions between 1200-1900 CE, with increased sea ice cover that resulted in insulation-induced warming between c. 1550-1750 CE (Harning et al., 2019). The inconsistent response in VGHV

temperatures to volcanic eruptions and solar minima between c. 1300-1800 CE could be associated with stochastic climate processes, such as the NAO, or a continued weakening of the SPG that counteracted the effects of negative radiative forcings and led to winter and spring warming over southern Iceland rather than cooling as observed in Northern Iceland.

On multi-decadal to centennial timescales, changes in the VGHV record do not consistently correspond to major temperature

anomalies observed during the summer months. The differences in seasonal climate responses to external forcings imply that the regional manifestation of these events depends on the initial state of the atmosphere and ocean but are also modulated by internal climate variability and changes in SSTs near southern Iceland associated with the SPG.

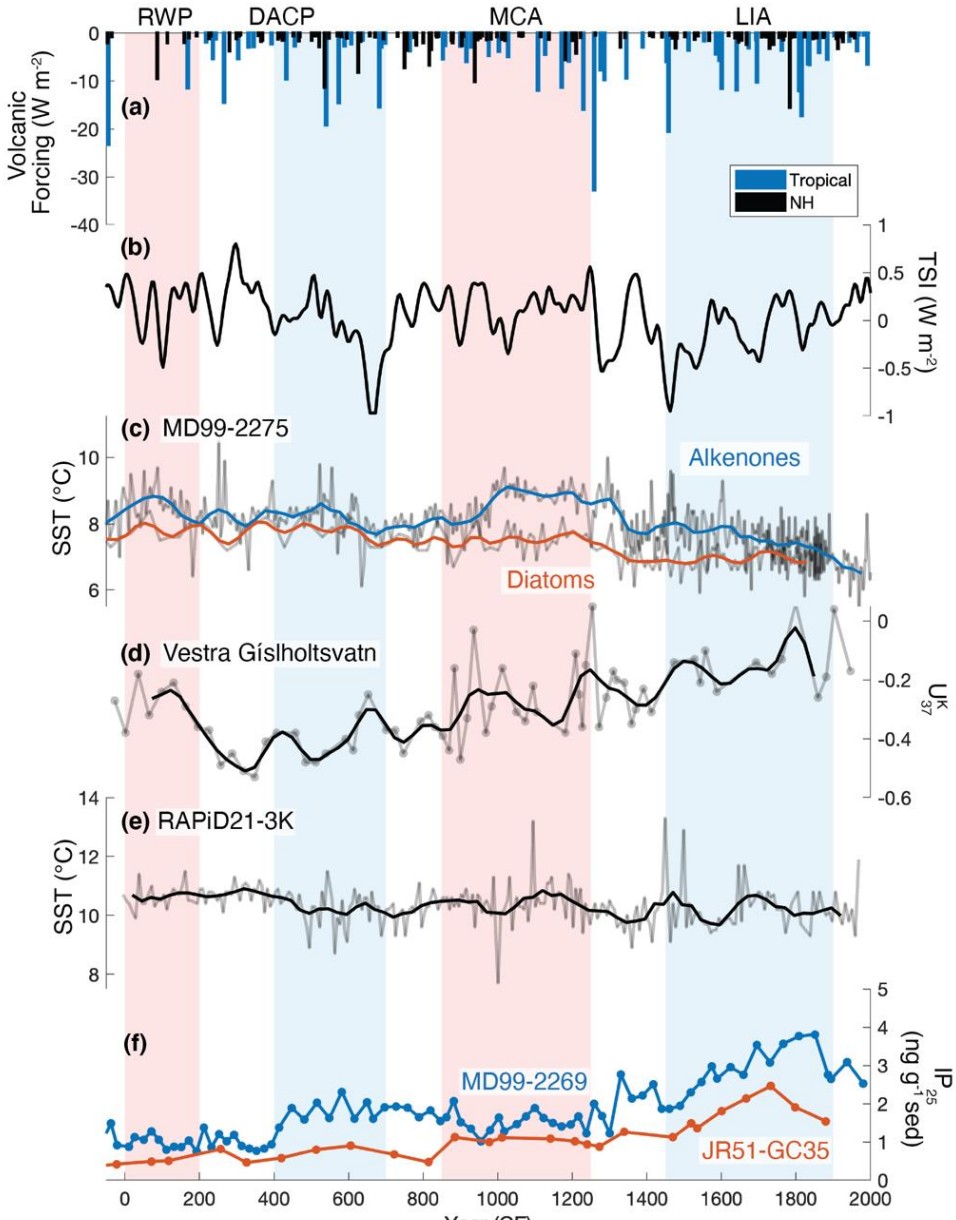

**Figure 7. Major changes in radiative forcings over the past 2,000 years, including (a) volcanic forcing from tropical and NH eruptions (Sigl et al., 2015) and (b) changes in total solar irradiance (TSI; Steinhilber et al., 2009) are shown. Marine and terrestrial reconstructions from Iceland are shown, including (c) alkenone and diatom sea surface temperature (SST) reconstructions from core MD99-2275 on the North Icelandic shelf (Sicre et al., 2008; Jiang et al., 2015), (d) an alkenone winter-spring temperature reconstruction from VGHV in southwest Iceland (this study), (e) an alkenone SST record from core RAPiD21-3K in the sub-polar North Atlantic (Sicre et al., 2011), and (f) sea-ice reconstructions developed using IP$_{25}$ from cores MD99-2269 (blue) and JR51-GC35 (orange) from the North Icelandic shelf (Cabedo-Sanz et al., 2016). The timing of major climate anomalies inferred from Icelandic climate records include: the Roman Warm Period (RWP, c. 0-200 CE), Dark Ages Cold Period (c. 400-700 CE), Medieval Climate Anomaly (MCA, c. 850-1250 CE), and Little Ice Age (LIA, c. 1450-1900 CE).**

## 5 Conclusions

The most striking feature of the VGHV record of winter-spring temperatures is a long-term warming trend from c. 250 CE to the present. Gradual warming in winter and spring temperatures is most likely associated with gradual warming of SSTs near Southern Iceland that are responding to increasing winter and spring solar insolation over the last 2,000 years. This contrasts with inferences of mean annual and summer time warming elsewhere in the NH. On multi-decadal to centennial timescales winter-spring temperatures are regional temperature responses to strong radiative forcings are often masked by internal climate

variability or changes in ocean circulation. These processes can cause regional differences in temperature and strong contrasts between winter and spring, summer, and mean annual temperatures. In general, this highlights a need for more winter and spring temperature reconstructions to improve our understanding of the magnitude and direction of cold-season temperature changes over the Common Era.

 **Appendix A**

**Table A1. Lake specific parameters for the model set up and references used to constrain the parameters.**

| Parameter | Value | Source |
|---|---|---|
| Obliquity | 23.4° | - |
| Latitude | 63.9°N | Blair et al. (2015) |
| Longitude | 23.9°W | Blair et al. (2015) |
| Local time (GMT) | +0 | - |
| Max. depth (m) | 15 | Blair et al. (2015) |
| Elevation of basin bottom (m.a.s.l) | 61 | Blair et al. (2015) |
| Catchment + Lake area (hectares) | 180 | Blair et al. (2015); Google Earth |
| Neutral drag coefficient | 0.002 | Longo et al. (2020) |
| Shortwave extinction coefficient (1/m) | 0.3 | Determined using $I_z/I_0 = e^{-\eta z}$ where $I_z/I_0 = 0.01$, $z = 15$ m, and $\eta$ is the extinction coefficient (Descy et al., 2006; |
| Fraction of advected air over lake | 0.3 | Dee et al., 2018; Longo et al. 2020) |
| Albedo of melting snow (slush) | 0.4* | Longo et al. (2020) |
| Albedo of non-melting snow | 0.7* | Longo et al. (2020) |
| Average depth (m) | 14 | Blair et al. (2015) |
| Salinity (ppt) | 0.0 | Longo et al. (2018) |

*Note: the albedo will be lower if there was a volcanic eruption that led to "dirty" snow and/or slush (Landl et al., 2003).*

**485** **Table A2. Outputs for lake model simulation and comparison with available observational data from Iceland.**

| Lake | Physical properties | | | | | Lake Surface Temp. (°C) | | Ice cover | | Citations |
|---|---|---|---|---|---|---|---|---|---|---|
| | Lat. (°N) | Lon. (°W) | Alt (m.a.sl.) | Max. Depth (m) | Area (km²) | Summer (JJA) | Winter (DJF) | Duration | Ice-out | |
| Simulation for VGHV | 63.9 | 23.9 | 61 | 14 | 1.57 | 9-10 | 0 | Dec.-Apr. | mid to late Apr. (avg. Apr. 23rd) | Blair et al. (2015), this study |
| Elliðavatn | 64.1 | 21.8 | 75 | 2.3 | 2.02 | 12-16 | 0-2 | N/A | N/A | Malmquist et al. (2009) |
| Mývatn | 65.6 | 17.0 | 278 | 4.2 | 53 | 8-13 | 0-3 | Oct.-May | May | Ólafsson (1979); Andradóttir (2012) |
| Thingvallavatn | 64.2 | 21.1 | 100.5 | 114 | 82 | 10-11 | 1-2 | Jan.-Apr. | Apr. (avg. Apr. 12th) | Adalsteinsson et al. (1992); Andradóttir (2012) |
| Skorarvatn | 66.3 | 22.3 | 183 | N/A | 1.92 | 8* | -2* | N/A | Apr.-May | Harning et al. (2020) |

*Note*: Skorarvatn temperatures are air temperatures that were based on a nearby meteorological station (Æðey) and were adjusted using the lapse rate for Skorarvatn.

**Table A3.** Ice-out dates determined using satellite imagery (Planet Labs Inc.). Due to cloud-cover we could not determine the exact dates of ice-off. We recorded the last day we observed ice on VGHV and the first day where we were able to observe VGHV as being completely ice-free.

| Year | Last day ice observed on lake | First day no ice observed on lake |
|---|---|---|
| 2012 | N/A | April 22 |
| 2014 | March 28 | April 13 |
| 2016 | March 17 | April 7 |
| 2018 | March 31 | April 23 |
| 2019 | March 30 | April 5 |
| **VGHV Simulated** | | **Average: April 23** |

**Table A4.** Results from the lake energy-balance model for air temperature perturbations showing the change in ice-off dates and lake water temperatures relative to the control simulation (average ice-off date: Apr. 23 ± 6 days, average lake water temperature = 4.1 ± 1.5 °C).

| Temperature Perturbation | Change in Ice-off Date (days) | Change in Surface Water Temperatures on May 1st (°C) |
|---|---|---|
| **DJF -7 °C** | 12.5 ± 8.3 | -2.0 ± 1.9 |
| **DJF -3 °C** | 4.9 ± 7.8 | -0.3 ± 2.0 |
| **DJF +3 °C** | -3.5 ± 9.3 | 0.1 ± 2.1 |
| **DJF +7 °C** | -3.8 ± 9.9 | 0.0 ± 2.1 |
| **MAM -7 °C** | 30.1 ± 7.8 | -3.1 ± 1.5 |
| **MAM -3 °C** | 13.8 ± 9.0 | -1.8 ± 2.0 |
| **MAM +3 °C** | -12.6 ± 9.9 | 1.6 ± 2.1 |
| **MAM +7 °C** | -25.9 ± 8.2 | 3.6 ± 2.2 |
| **JJA -7 °C** | 0.2 ± 8.1 | 0.0 ± 2.2 |
| **JJA -3 °C** | 0.2 ± 8.1 | 0.0 ± 2.2 |
| **JJA +3 °C** | 0.0 ± 8.2 | 0.0 ± 2.2 |
| **JJA +7 °C** | 0.0 ± 8.2 | 0.0 ± 2.2 |
| **SON -7 °C** | 0.8 ± 8.0 | 0.5 ± 2.0 |
| **SON -3 °C** | 0.9 ± 8.8 | 0.2 ± 2.1 |
| **SON +3 °C** | 0.6 ± 8.7 | -0.2 ± 2.1 |
| **SON +7 °C** | 0.4 ± 8.7 | 0.0 ± 2.1 |

**Table A5. Results from the lake energy-balance model for shortwave radiation perturbations showing the change in ice-off dates and lake water temperatures relative to the control simulation (average ice-off date: Apr. 23 ± 6 days, average lake water temperature = 4.1 ± 1.5 °C).**

| Shortwave Solar Radiation Perturbation | Change in Ice-off Date (days) | Change in Surface Water Temperatures on May 1st (°C) |
|---|---|---|
| DJF - 4 W m$^{-2}$ | 0.3 ± 8.1 | 0.0 ± 2.2 |
| DJF - 2W m$^{-2}$ | 0.2 ± 8.1 | 0.0 ± 2.2 |
| DJF + 2 W m$^{-2}$ | -0.2 ± 8.2 | 0.0 ± 2.2 |
| DJF + 4 W m$^{-2}$ | -0.5 ± 8.3 | 0.0 ± 2.2 |
| MAM - 4 W m$^{-2}$ | 1.0 ± 8.4 | 0.0 ± 2.1 |
| MAM - 2 W m$^{-2}$ | 0.3 ± 8.1 | 0.0 ± 2.2 |
| MAM + 2 W m$^{-2}$ | -0.3 ± 8.2 | 0.0 ± 2.2 |
| MAM + 4 W m$^{-2}$ | -0.7 ± 8.2 | 0.1 ± 2.2 |
| JJA - 4 W m$^{-2}$ | 0.0 ± 8.2 | 0.0 ± 2.2 |
| JJA - 2 W m$^{-2}$ | 0.0 ± 8.2 | 0.0 ± 2.2 |
| JJA + 2 W m$^{-2}$ | 0.0 ± 8.2 | 0.0 ± 2.2 |
| JJA + 4 W m$^{-2}$ | 0.0 ± 8.2 | 0.0 ± 2.2 |
| SON - 4 W m$^{-2}$ | 0.2 ± 8.1 | 0.0 ± 2.2 |
| SON - 2 W m$^{-2}$ | 0.2 ± 8.1 | 0.0 ± 2.2 |
| SON + 2 W m$^{-2}$ | -0.1 ± 8.2 | 0.0 ± 2.2 |
| SON + 4 W m$^{-2}$ | -0.2 ± 8.2 | 0.0 ± 2.2 |

 **Table A6. Results from the lake energy-balance model for longwave radiation perturbations showing the change in ice-off dates and lake water temperatures relative to the control simulation (average ice-off date: Apr. 23 ± 6 days, average lake water temperature = 4.1 ± 1.5 °C).**

| Longwave Radiation Perturbation | Change in Ice-off Date (days) | Change in Surface Water Temperatures on May 1st (°C) |
|---|---|---|
| -0.2 W m$^{-2}$ | 0.2 ± 8.1 | 0.0 ± 2.2 |
| +0.2 W m$^{-2}$ | 0.0 ± 8.2 | 0.0 ± 2.2 |

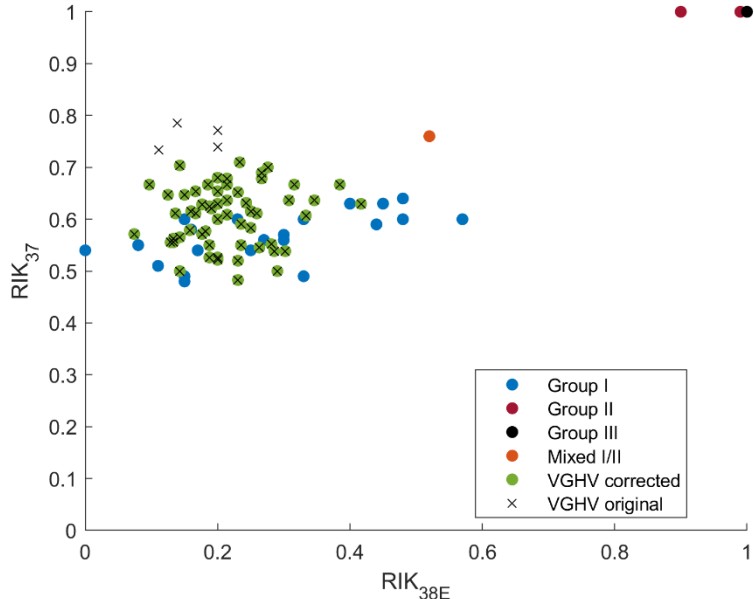


**Figure A1. Comparison of RIK$_{37}$ and RIK$_{38E}$ values for different alkenone distributions. Group I alkenone values (blue) and a mixed sample containing Group I and II (orange) are from the Northern Hemisphere and Northern Alaska datasets (Longo et al., 2016, 2018). The Group II and III values were determined from cultures (see Longo et al. 2018). The original VGHV alkenone distributions are indicated by an X (these points only include samples where we were**

**able to reliably identify and quantify C$_{38:3}$Et and its isomer). The VGHV alkenones that were corrected for Group II are shown in green.**

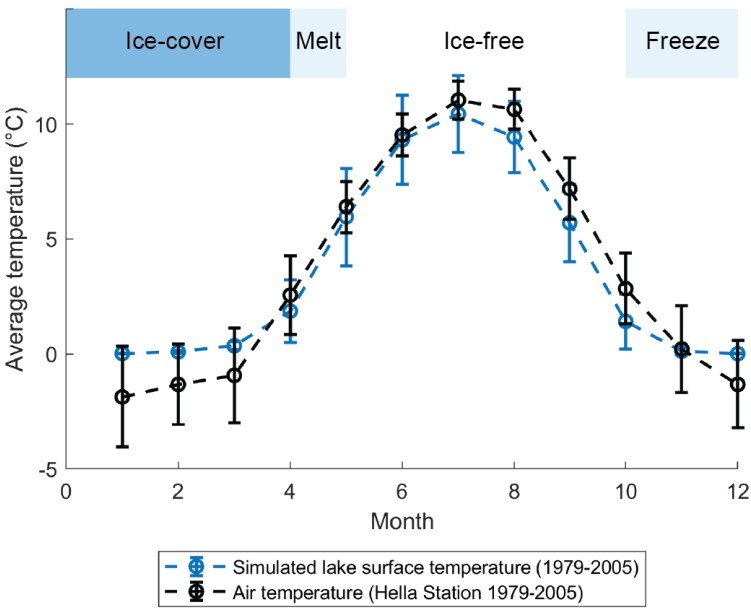

**Figure A2. Comparison of simulated lake surface water temperatures for VGHV (blue) and air temperatures from a nearby meteorological station (Hella Station) for the period of 1979-2005. The average duration of ice-cover simulated in our lake model is shown in dark blue. The average timing of ice-off dates and the timing of when lake ice-cover starts to form that is simulated in our lake model are shown in pale blue.**


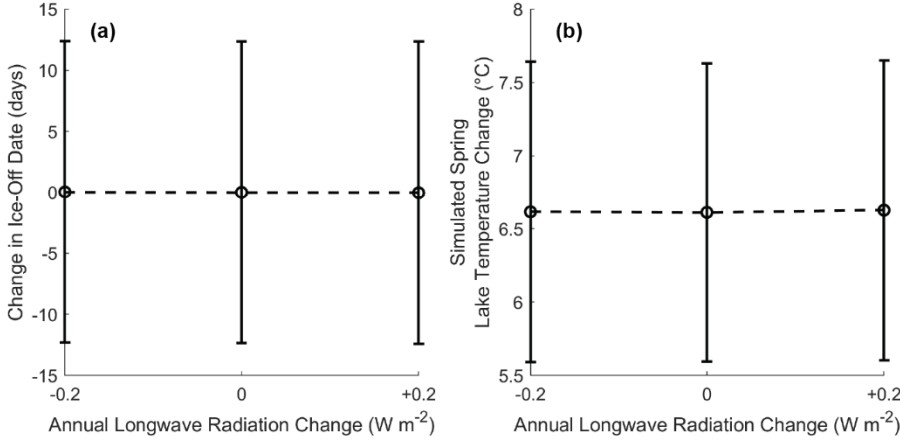


**Figure A3. Lake model sensitivity tests showing the effect of annual longwave radiation perturbations on (a) ice-off dates and (b) lake water temperatures on May 1st.**

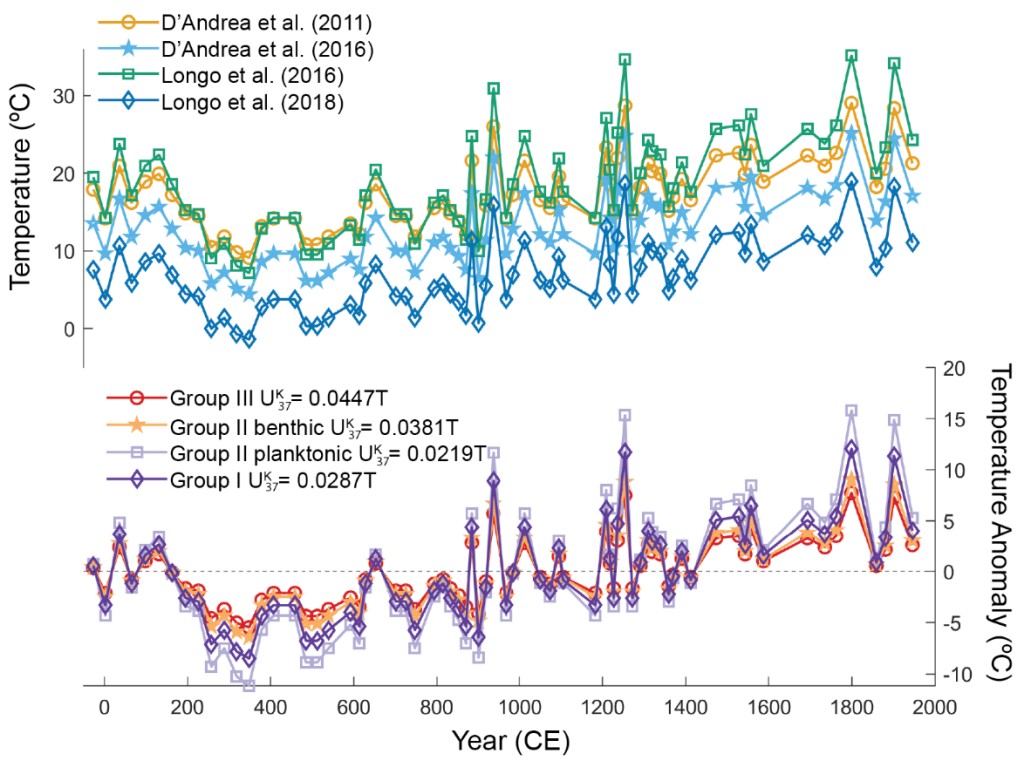

**Figure A4. Alkenone calibrations from previous studies including (a) $U_{37}^K = 0.0296T - 0.80$ (D'Andrea et al., 2011), $U_{37}^K = 0.0284T - 0.655$ (D'Andrea et al., 2016), $U_{37}^K = 0.021T - 0.68$ (Longo et al., 2016), and $U_{37}^K = 0.029T - 0.49$ (Longo et al., 2018). (b) Temperature anomalies calculated from slopes previously determined by D'Andrea et al., (2016) for Group III, Group II benthic, Group II planktonic, and Group I.**


**Data availability**

Data will be made available at the National Oceanic and Atmospheric Administration National Centers for Environmental Information (NOAA NCEI) Paleoclimate Database: https://www.ncdc.noaa.gov/paleo/study/29992. The age model and information about the lake sediment core were obtained from Blair et al. (2015). Information about the lake energy balance

model used in this study can be found in Dee et al. (2018) and the code for the lake energy balance model is available at: https://github.com/sylvia-dee/PRYSM. ERA-Interim daily data (1979-2019 CE) was obtained from: https://www.ecmwf.int/en/forecasts/datasets/reanalysis-datasets/era-interim (ECMWF; Dee et al., 2011). Meteorological data for southwest Iceland was obtained from: (https://en.vedur.is/climatology/data/; Icelandic Meteorological Office). Data used to make the maps in Fig. 1 can be found at: Natural Earth (https://www.naturalearthdata.com/), the National Land Survey of

Iceland (https://www.lmi.is/en/), and the National Oceanic and Atmospheric Administration (NOAA) World Ocean Database (https://www.nodc.noaa.gov/OC5/WOD/pr_wod.html; Boyer et al., 2018). Satellite imagery for Table A3 was obtained from Planet Labs Inc. (https://www.planet.com/).

**Author contributions**

Study conceptualized by NR, JMR, and YH. Method development and laboratory analyses by NR and JG. NR prepared the

manuscript with contributions from all co-authors.

**Competing interests**

The authors declare that they have no conflict of interest.

**Acknowledgements**

This project was funded by Geological Society of America Graduate Student Research Grants, the Nicole Rosenthal Hartnett

Graduate Fellowship, Brown University Graduate School, and Brown University Undergraduate Teaching and Research Awards. We would like to thank Prof. T.D. Herbert, R. Rose, Dr. J. Salacup, Dr. G. Weiss, Dr. C. Morrill, A. Neary, Prof. S.G. Dee, and E. Kyzivat for advice and analytical support. All of the samples for this project were obtained from LacCore (National Lacustrine Core Facility), Department of Earth Sciences, University of Minnesota-Twin Cities. We would also like to thank the reviewers and David Harning for helping to improve this manuscript.

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
