# Peer review of "Winter-spring warming in the North Atlantic during the last 2,000 years: Evidence from Southwest Iceland"

_Climate of the Past, 2020_

## Referee Comment (RC1) · Anonymous Referee #1 · 13 Aug 2020

Richter et al. present a temperature reconstruction for a season that is rarely captured by proxy archives. To do so, they apply an exciting emerging alkenone-based proxy (UK37) using an elaborate extraction and purification procedure. The authors build on a previously published robust chronological framework that allows them to confidently resolve shifts in multi-decadal climate conditions. I do, however, have a number of major concerns about the analysis and interpretation of this work:

The presented dataset suffers from species mixing, complicating its interpretation as a temperature record. Now, the authors use RIK37 cut-off values to exclude samples that are dominated by Group II haptophytes. As calibrations exist for this phylotype,

a significant portion of their data points is excluded as a consequences of this rather crude solution. I would recommend the authors to calculate the RIK38E index to better differentiate phylogeny (mixing) and derive temperatures from Group II data.

I commend the authors for their efforts to better constrain the seasonality of hapto-phyte production (and temperature sensitivity), but have two concerns. First, ice-off dictates the timing of haptophyte blooms: with this in mind, I wonder why the authors did not rely on satellite data to validate the 30 yr control run outlined in section 2.4. High-res imagery is freely available for the entire period: if in agreement with model output, this would significantly strengthen the robustness of their approach. Secondly, the presented modelling results reveal that both late winter as well as spring season temperatures help determine ice-off dates: this does not justify presenting the record as a "cod season" reconstruction.

See line 215: I think the wording is far too strong here. The authors argue existing calibrations provide "unreasonable" estimates and back this up with unrealistically high temperature values. They do, however, not state that these values were calculated using site-specific intercepts provided for each of the used calibrations while the au-thors of the applied calibrations advise against doing so. To remedy this, I advise the authors to discuss the relative temperature fluctuations plotted in Fig. A2(b): indeed, the magnitude of these swings are of equal magnitude as those observed during the spring transitional season (Fig. 5b).

Paragraph around line 230: here the authors try to relate their reconstructions to warm-ing/cooling periods that are often referenced in the (North Atlantic) literature. I would stay clear from this and consider removing this section for a number of reasons. First, a string of recent studies has underlined just how spatio-temporally heterogeneous ex-pression of these events is (see e.g. Werner et al. 2018 – COP, McKay et al. 2018 – GRL, and van der Bilt et al. 2019 – QSR). Secondly, most of these events are most clearly expressed in summer, while the authors argue that their record captures "cold season" conditions. Finally, and related to this, the perceived correspondence is tenuous at best as the authors also confirm by using wording like "roughly coincides" or "could be associated".

Section 4.1: please restructure and tighten this paragraph. As the presented record only covers the past 2millennia, I think the current full Holocene focus is not the right way to frame things. Also, the authors allude to the so-called "Holocene temperature Conundrum" but don's state so (or explain it clearly). The way I see things, the main message here is that spring temperatures are (not entirely surprisingly) not driven by changes in summer insolation. I would contextualize/strengthen this by discussing other non-summer temperature reconstructions (which the authors already do to some extent), and argue why one would expect to see this "cold season" imprint in a maritime Arctic setting like Iceland, where it is known that many feedbacks may overprint any radiative signature, notably surface ocean currents, but also sea-ice feedbacks – in this respect, I recommend the authors to check Park et al. 2019 – Science Advances. Finally, as the authors point out in section 2.4, Iceland receives little sunlight during winter: I therefore recommend them to plot early spring insolation in Fig. 6a instead of winter + spring insolation.

Section 4.2: the authors (partly) attribute higher-frequency changes to shifts in regional climate dynamics, notably the NAO. When doing so, it would be most helpful to provide a contextual understanding of this complex system on Iceland – what happens to the different components of the regional climate system during (shifts between) positive/negative NAO phases. Now, it oft feels as if this discussion is shoehorned into an NAO mould using a hotchpotch of sources. Also, respect the sampling and chronological resolution of this dataset: I don't think it warrants attribution to multi-annual forcing mechanisms.

---

## Short Comment (SC1) · 20 Aug 2020

Richter et al. present an interesting alkenone-derived proxy dataset, which is nicely paired with a lake energy balance model to help determine proxy seasonality. However, in light of other work indicating the spring/summer seasonality of alkenones in Iceland and the known human impact on the Icelandic landscape during the last 2 ka, I think there is room for some of the text (mostly discussion) to be revised.

Alkenone seasonality: I find it difficult to rationalize the cold season bias for alkenone seasonality in VGHV given 1) the minor changes NH winter insolation anomalies (and minimal winter sunlight in Iceland) relative to NH summer insolation, and 2) that

alkenone Uk'37 values in a Holocene lake record from NW Iceland track the first-order decrease in NH summer insolation (Harning et al., 2020). I think it would be valuable to discuss the differences between the NW and S Iceland alkenone records in terms of their relation to seasonal insolation values and what this may mean for their interpretation. Reviewer 1 mentions exploring satellite imagery as well, which I second, as it may reveal differences in the timing of ice out between NW and S Iceland that may account for some of the differences. Additionally, the authors could compare against the long instrumental temperature record from Iceland, which has been used to show that biogenic silica in a west Iceland lake (Haukadalsvatn) correlates best with April/May temperatures (Geirsdóttir et al., 2009).

Regarding the model, I imagine summer insolation values also influence the timing of ice out. If summer insolation is decreasing, could this lead to more persistent lake ice over VGHV? Several studies on North Iceland marine SST proxy records, including alkenones, have noted a similar "warming" over the last 2 ka (e.g. Moossen et al., 2015; Kristjánsdóttir et al., 2017), possibly as a function of sea ice lasting longer into the spring/summer, resulting in algal blooms occurring later in the season during relatively warmer months (e.g. Cabedo-Sanz et al., 2016). Could this possibly explain some degree of the warming observed in VGHV? If the record does indeed reflect winter warming, I'd suggest also comparing with a relatively new winter subsurface temperature record from the North Iceland Shelf (Harning et al., 2019).

Landscape disturbances: The authors mention several times that VGHV's alkenone record is not affected by additional confounding variables (e.g. L44-46, L270-273). However, culture experiments from other haptophyte species (which admittedly are not the biological source of those in VGHV) show that Uk'37 values are sensitive to additional factors, such as nutrients and light (e.g. Prahl et al., 2003). From the existing proxy records from VGHV we know that human settlement in the catchment beginning at ∼1.1 ka BP had a significant impact on the surrounding landscape, including changes in vegetation (pollen) and an increased contribution of terrestrial material to

the lake (C/N) (Blair et al., 2015). Presumably these changes could also have delivered more nutrients to the lake, so could it be possible some of the changes in alkenone unsaturation (and/or RIK37 values, L164-166) are also influenced by human-settlement-related disturbances? In addition, tephra fall can also destabilize the landscape (Larsen et al., 2011; Eddudottir et al., 2017) so I wonder if VGHV's proximal location to the active volcanic zone and the heavy tephra loading during the last 2 ka may also impact nutrient flux and associated changes in Uk'37. In any regard, I think exploring the relationship of the new alkenone record with the existing proxy records (Blair et al., 2015) would be a valuable addition to the manuscript. As the record stands right now without the full Holocene for perspective of natural climate variability, it's difficult to confidently assign the changes observed to simply climate rather than some combination of that and human disturbances.

References

Blair, C.L., Geirsdóttir, AÌĄ., and Miller, G.H.: A high-resolution multi-proxy lake record of Holocene environmental change in southern Iceland. J. Quat. Sci., 30, 281-292, 2015.

Cabedo-Sanz, P., Belt, S.T., Jennings, A.E., Andrews, J.T., and Geirsdóttir, Á.: Variability in drift ice export from the Arctic Ocean to the North Icelandic Shelf over the last 8000 years: A multi-proxy evaluation. Quat. Sci. Rev., 146, 99-115, 2016.

Eddudóttir, S.D., Erlendsson, E., Gísladóttir, G.: Effects of the Hekla 4 tephra on vegetation in Northwest Iceland. Veg. Hist. Archaeobot., 26, 389-402, 2017.

Geirsdóttir, A., Miller, G.H., Thordarson, T., Ólafsdóttir, S.: A 2000 year record of climate variations reconstructed from Haukadalsvatn, West Iceland. J. Paleolimnol., 41, 95-115, 2009.

Harning, D.J., Andrews, J.T., Belt, S.T., Cabedo-Sanz, P., Geirsdóttir, Á., Dildar, N., Miller, G.H., and Sepúlveda, J.: Sea ice control on winter subsurface temperatures

of the North Iceland Shelf during the Little Ice Age: A TEX86 calibration case study. Paleoceanogr. Paleoclimatol., 34, 1006-1021, 2019.

Harning, D.J., Curtin, L., Geirsdóttir, Á., D'Andrea, W.J., Miller, G.H., and Sepúlveda, J.: Lipid biomarkers quantify Holocene summer temperature and ice cap sensitivity in Icelandic lakes. Geophys. Res. Lett., 47, e2019GL085728, 2020.

Kristjánsdóttir, G.B., Moros, M., Andrews, J.T., Jennings, A.E.: Holocene Mg/Ca, alkenones, and light stable isotope measurements on the outer North Iceland shelf (MD99-2269): A comparison with other multi-proxy data and sub-division of the Holocene. Holocene, 26, 55-62, 2017.

Larsen, D.J., Miller, G.H., Geirsdóttir, Á., Thordarson, T.: A 3000-year varved record of glacier activity and climate change from the proglacial lake Hvítárvatn, Iceland. Quat. Sci. Rev., 30, 2715-2731, 2011.

Moossen, H., Bendle, J., Seki, O., Quillmann, U., Kawamura, K.: North Atlantic Holocene climate evolution recorded by high-resolution terrestrial and marine biomarker records. Quat. Sci. Rev., 129, 111-127, 2015.

Prahl, F.G., Wolfe, G.V., and Sparrow, M.A.: Physiological impacts on alkenone paleothermometry. Paleoceanography, 18, 1-7, 2003.

---

## Referee Comment (RC2) · Anonymous Referee #2 · 29 Sep 2020

Richter et al. reconstruct cold-season temperature trends over the last 2 ky for southwest Iceland using alkenones produced by lacustrine haptophyte algae. The authors demonstrate that alkenones from lake VGHV record a long-term warming trend, as well as decadal to centennial scale variability within the long-term trend. They couple this temperature reconstruction with a lake energy balance model to support that increasing high-latitude winter insolation is likely responsible for the overarching cold-season warming for the last ~2 ky, while climate perturbations are likely responsible for high frequency variability in proxy data. The authors contextualize this data in a broader framework by suggesting that this dataset, and more studies like it, could help consolidate discrepancies between global climate model output and proxy reconstructions for

the northern hemisphere through the Holocene.

Major contributions of this manuscript:

This work offers important insight into seasonal differences in temperature for SW Iceland (and by inference this part of the N Atlantic) for the last 2 ky

The coupling of proxy inference and lake energy model is a progressive approach for interpreting proxy data by testing it within varying climate forcing scenarios

The presentation of the data is thoughtful and clear

Criticisms of the manuscript:

I find the discussion around the seasonality of this proxy, and the conclusions drawn from it, to be somewhat confusing and at times inconsistent. I think it would be helpful if the manuscript more clearly articulated the chain of logic/evidence that provides that alkenones, which are stated to bloom in spring, can be interpreted more broadly as a record of cold-season temperatures driven by cold-season insolation.

Within this point, I would find it helpful if the background discussion around the proxy touched on the fidelity of alkenones for reconstructing temperature (is it known to have significant error associated with it, or low significance values?) and are there calibration data that covers a climatically similar region? I appreciate that this record is being interpreted qualitatively and it is clearly stated in the manuscript that there is no local calibration data, but I think it would improve confidence in this interpretation of the data to know that it has been tested/utilized in comparable locations, particularly in interpreting high frequency changes as related to climate perturbations and not stochastic proxy noise

I would find it valuable to know if there is a competing effect from declining summer insolation/temperature on spring temperatures and the timing of ice-out. It's unclear if JJA/SON is held constant in the model, or if there is little response to ice-out date/water temps given changing temps/insolation during these seasons (Fig. 5).

Is the lake model consistent with observational data for what controls lake ice-out dates & water temps? (i.e. are there examples in modern observational data of earlier ice-off dates in regions with increasing winter air temperatures?)

Some discussion around if the parameters & outputs of this model are climatically probable for the coverage of the record would improve this manuscript. E.g. An air temperature increase in the winter of +7 deg does dramatically move the ice-out date, but that change in temperature seems far too large for the amount of cold-season insolation change. These bounds, I think, are justified in lines 145-157, but I find this statement/constraint confusing. Observed temperatures in any given season can range by +/- 7, but why would average cold-season temperatures range by this amount over the last 2 ky? Change in insolation seems to have much less impact in ice-out dates in the model (Fig 5) but is credited for driving trends in alkenone data. This reads as a mismatch in results-conclusions as written, and the manuscript would improve from some text that consolidates the results of the model with their interpretation of the data.

The warming trend apparent in this data really seems to start just before ∼400 CE, with temperature values only returning to average values from the start of the record several centuries later. This could indicate the long-term trend is less pronounced than what is captured by this window (i.e. there is a rebounding from depressed temperatures, and then warming above that average only over the last millennia). I think it would be an interesting point to add to consider if the early values (0-200 CE) are the anomaly of the record, of if the record should be considered in the context of these early values.

Overall I think this is a significant and important study, but the manuscript would benefit from some additions to background information and from adding text that consolidates what is learned in the model with their proxy data, and the climate implications of these data, so that it is very clear the conclusions are supported by their data prior to publication.

---

## Author Comment (AC2) · 4 Nov 2020

**Response to David Harning**

We would like to thank David Harning for taking the time to read our manuscript and for posting a comment. We have included the comments below along with our responses.

*Comment:* Richter et al. present an interesting alkenone-derived proxy dataset, which is nicely paired with a lake energy balance model to help determine proxy seasonality. However, in light of other work indicating the spring/summer seasonality of alkenones in Iceland and the known human impact on the Icelandic landscape during the last 2 ka, I think there is room for some of the text (mostly discussion) to be revised.

Alkenone seasonality: I find it difficult to rationalize the cold season bias for alkenone seasonality in VGHV given 1) the minor changes NH winter insolation anomalies (and minimal winter sunlight in Iceland) relative to NH summer insolation, and 2) that alkenone Uk'37 values in a Holocene lake record from NW Iceland track the first-order decrease in NH summer insolation (Harning et al., 2020). I think it would be valuable to discuss the differences between the NW and S Iceland alkenone records in terms of their relation to seasonal insolation values and what this may mean for their interpretation. Reviewer 1 mentions exploring satellite imagery as well, which I second, as it may reveal differences in the timing of ice out between NW and S Iceland that may account for some of the differences. Additionally, the authors could compare against the long instrumental temperature record from Iceland, which has been used to show that biogenic silica in a west Iceland lake (Haukadalsvatn) correlates best with April/May temperatures (Geirsdóttir et al., 2009).

*Response:* As demonstrated in our lake energy balance model and previous work by Longo et al. (2016, 2018), lake water temperatures during the main period of alkenone production (i.e. ice-off) are sensitive to air temperatures during the winter and spring. Results from Longo et al. (2020) also show that this long-term trend in winter-spring insolation recorded by Group I alkenones holds true throughout the Holocene in Alaska. Unfortunately, the low-resolution alkenone record (resolution c. 1,000 years) in Harning et al. (2020) make it difficult to resolve whether the record is primarily recording spring or summer insolation during the last 2000 years, which overlaps with our record. Further, ice-off dates in Skorarvatn (late April-May, Harning et al., 2020) occur later than ice-off dates in VGHV (early to mid-April) due to the colder temperatures and sea-ice formation off the coast of Northern Iceland. As described above, this may account for the difference in seasonal sensitivity. The differences in the interpretation of alkenone records from Iceland, highlights the need for an *in situ* study of alkenone production in Icelandic lakes. In line with your suggestion, we expanded on the comparisons between northern and southern Iceland in our discussion, section 4.1.

     As discussed in our response to reviewer 1, the purpose of the lake model is to determine the sensitivity of lake water temperatures and ice-melt to different processes. Further, the alkenone bloom actually starts prior to when the lake is completely ice free (Longo et al., 2018), and therefore knowing the exact date of ice-off will not alter our conclusions. In terms of comparing ice-off dates between NW and S Iceland, we agree that this might be interesting to pursue in future studies.

*Comment:* Regarding the model, I imagine summer insolation values also influence the timing of ice out. If summer insolation is decreasing, could this lead to more persistent lake ice over VGHV? Several studies on North Iceland marine SST proxy records, including alkenones, have noted a similar "warming" over the last 2 ka (e.g. Moossen et al., 2015;

Kristjánsdóttir et al., 2017), possibly as a function of sea ice lasting longer into the spring/summer, resulting in algal blooms occurring later in the season during relatively warmer months (e.g. Cabedo-Sanz et al., 2016). Could this possibly explain some degree of the warming observed in VGHV? If the record does indeed reflect winter warming, I'd suggest also comparing with a relatively new winter subsurface temperature record from the North Iceland Shelf (Harning et al., 2019).

*Response:* As demonstrated in our lake model and previous work by Longo et al. (2020) in Alaska, decreasing summer insolation does not result in more persistent lake ice cover and most of the changes in lake ice cover are in fact driven by temperature changes during the winter and spring season. As discussed in our response to reviewer 2, VGHV freezes every winter meaning that the minimal lake water temperature is reached and the lake water temperatures are essentially "reset."

We will consider discussing the Harning et al. (2019) winter subsurface temperature record in our modified text (section 4.1). As noted in the previous comment the presence of sea-ice and effects of the East Greenland Current may have a stronger influence on the subsurface temperature record from the North Icelandic Shelf, which might contribute to differences observed between North and South Iceland.

*Comment:* Landscape disturbances: The authors mention several times that VGHV's alkenone record is not affected by additional confounding variables (e.g. L44-46, L270-273). However, culture experiments from other haptophyte species (which admittedly are not the biological source of those in VGHV) show that Uk'37 values are sensitive to additional factors, such as nutrients and light (e.g. Prahl et al., 2003). From the existing proxy records from VGHV we know that human settlement in the catchment beginning at _1.1 ka BP had a significant impact on the surrounding landscape, including changes in vegetation (pollen) and an increased contribution of terrestrial material to the lake (C/N) (Blair et al., 2015). Presumably these changes could also have delivered more nutrients to the lake, so could it be possible some of the changes in alkenone unsaturation (and/or RIK37 values, L164-166) are also influenced by human-settlement related disturbances? In addition, tephra fall can also destabilize the landscape (Larsen et al., 2011; Eddudottir et al., 2017) so I wonder if VGHV's proximal location to the active volcanic zone and the heavy tephra loading during the last 2 ka may also impact nutrient flux and associated changes in Uk'37. In any regard, I think exploring the relationship of the new alkenone record with the existing proxy records (Blair et al., 2015) would be a valuable addition to the manuscript. As the record stands right now without the full Holocene for perspective of natural climate variability, it's difficult to confidently assign the changes observed to simply climate rather than some combination of that and human disturbances.

*Response:* We have a separate paper discussing the impacts of environmental disturbances on the lake sediment record and how we disentangle human impacts from climate related changes (Richter at al., *in review*). To summarize our main conclusions from this paper and as briefly discussed in section 3.1, we are able to isolate the climate influenced signal from human impacts by removing data points that are influenced by Group II Isochrysidales. In lakes, changes in species composition (i.e. Group I vs. Group II), and therefore alkenone production, are primarily influenced by salinity and trophic conditions (Longo et al., 2018; Yao et al., 2019; Plancq et al., 2018, 2019). As major salinity changes are highly unlikely to have occurred in VGHV over the past 2,000 years, we assume that most of the changes in species composition are driven by a change in nutrient inputs. An increase in $RIK_{37}$ values, and therefore Group II Isochrysidales, occurs after 874 CE, around the time early Norse

settlers arrived in Iceland. We independently constrain these early human impacts using long-chain *n*-alkanes to track major changes in vegetation composition (Richter et al., *in review*). The timing of human impacts on VGHV is very well-constrained, and although landscape and other changes did impact the haptophyte community, after correcting for species mixing we observe no large shifts in our temperature record at 874 CE.

Unfortunately, there are no available cultures from Group I to validate that Group I alkenone production is not influenced by changes in nutrient conditions or light. However, as demonstrated by previous studies, alkenone production by Group I Isochrysidales is primarily influenced by changes in spring lake water temperatures (Longo et al., 2016, 2018; Plancq et al., 2018, 2019; Yao et al., 2019). In terms of volcanic tephra layers, we see no correlation between changes in our alkenone record (either $RIK_{37}$ or $U_{37}^{K}$) and tephra layers identified in the VGHV sediment core over the past 2,000 years.

**References**

Harning, D.J., Andrews, J.T., Belt, S.T., Cabedo-Sanz, P., Geirsdóttir, Á., Dildar, N., Miller, G.H. and Sepúlveda, J.: Sea Ice Control on Winter Subsurface Temperatures of the North Iceland Shelf During the Little Ice Age: A TEX86 Calibration Case Study. Paleoceanogr. Paleoclimatol., 34, 1006-1021, 2019.

Harning, D. J., Curtin, L., Geirsdóttir, Á., D'Andrea, W. J., Miller, G. H., and Sepúlveda, J.: Lipid biomarkers quantify Holocene summer temperature and ice cap sensitivity in Icelandic lakes. Geophys. Res. Lett., 47, 2020.

Longo, W. M., Theroux, S., Giblin, A. E., Zheng, Y., Dillon, J. T., and Huang, Y.: Temperature calibration and phylogenetically distinct distributions for freshwater alkenones: evidence from northern Alaskan lakes. Geochim. Cosmochim. Ac., 180, 177-196, https://doi.org/10.1016/j.gca.2016.02.019, 2016.

Longo, W.M., Huang, Y., Yao, Y., Zhao, J., Giblin, A.E., Wang, X., Zech, R., Haberzettl, T., Jardillier, L., Toney, J., and Liu, Z.: Widespread occurrence of distinct alkenones from Group I haptophytes in freshwater lakes: Implications for paleotemperature and paleoenvironmental reconstructions. Earth Planet. Sci. Lett., 492, 239-250, https://doi.org/10.1016/j.epsl.2018.04.002, 2018.

Longo, W. M., Huang, Y., Russell, J. M., Morrill, C., Daniels, W. C., Giblin, A. E., and Crowther, J.: Insolation and greenhouse gases drove Holocene winter and spring warming in Arctic Alaska. Quat. Sci. Rev., 242, 2020.

Plancq, J., Cavazzin, B., Juggins, S., Haig, H. A., Leavitt, P. R., and Toney, J. L.: Assessing environmental controls on the distribution of long-chain alkenones in the Canadian Prairies. Org. Geochem., 117, 43-55, 2018.

Plancq, J., Couto, J. M., Ijaz, U. Z., Leavitt, P. R., and Toney, J. L.: Next-Generation Sequencing to Identify Lacustrine Haptophytes in the Canadian Prairies: Significance for Temperature Proxy Applications. J. Geophys. Res Biogeosci, 124, 2144-2158, 2019.

Richter, N., Russell, J.M., Garfinkel, J., and Huang, Y.: Impacts of Norse settlement on terrestrial and aquatic ecosystems in Southwest Iceland. In review at the Journal of Paleolimnology.

Yao, Y., Zhao, J., Longo, W.M., Li, G., Wang, X., Vachula, R.S., Wang, K.J. and Huang, Y.: New insights into environmental controls on the occurrence and abundance of Group I alkenones and their paleoclimate applications: Evidence from volcanic lakes of northeastern China. Earth Planet. Sci. Lett., 527, 2019.

---

## Author Response (AR1)

Dear Bjørg Risebrobakken,

Thank you for taking the time to review our manuscript, we hope the edits that we outline below and that we made in the manuscript will address the comments.

Sincerely,
Nora Richter

***Editor comment:*** Out of curiosity from reading your responses: You state e.g. that while you will discuss the relation to existing SST data, including the subsurface data by Harning, you are unsure of the relevance of subsurface temperatures. From a marine point of view I would find this comparison interesting with respect to your seasonality discussion, expecting the annual mean/winter/spring temperature development of the surface water to be reflected by the subsurface record. At least in the eastern Nordic Seas we see that the subsurface temperature is constant year round, comparable to the winter/spring surface temperatures (e.g. Nyland et al., 2006; Jansen et al., 2008; Andersson et al., 2010; Risebrobakken et al., 2011). Hence, I wonder if it is possible to use existing information with respect to seasonal responses from nearby marine records to strengthen your argumentation for seasonality impact on your records?

***Response:*** Thank you for the comment. We included a discussion on nearby marine records and the seasonal responses in our manuscript (see section 4.1):

"SST reconstructions near southern Iceland show that surface temperatures either increased (Berner et al., 2008; Thornalley et al., 2009; Miettinen et al., 2012; Orme et al., 2018) or did not significantly change (Sicre et al., 2011; Van Nieuwenhove et al., 2018) over the last 2,000 years. Marine reconstructions of temperature from below the summer thermocline and bottom water record a decrease in mean annual temperatures over the Common Era (Thornalley et al., 2009; Ólafsdóttir et al., 2010; Moffa-Sánchez et al., 2014) as the transport of warm North Atlantic Current waters by the Irminger Current decreased over the last 2,000 years (Ólafsdóttir et al., 2010). Based on existing paleo- and historical records we conclude that sea ice feedbacks only play a minor role in driving long-term changes in winter and spring temperatures at our study site, whereas an increase in SSTs along the southern coast could contribute to the warming trend observed in our record. However, discrepancies in existing proxy records makes it difficult to correlate changes in SSTs to changes in winter and spring temperatures at VGHV." (lines 292-301)

**Response to Anonymous Referee #1**

We would like to thank the referee for taking the time to review our manuscript and for their comments. We have included the reviewer comments below along with our responses and edits to the manuscript (highlighted in blue).

*Reviewer comment:* Richter et al. present a temperature reconstruction for a season that is rarely captured by proxy archives. To do so, they apply an exciting emerging alkenone-based proxy (UK37) using an elaborate extraction and purification procedure. The authors build on a previously published robust chronological framework that allows them to confidently resolve shifts in multi-decadal climate conditions. I do, however, have a number of major concerns about the analysis and interpretation of this work:

The presented dataset suffers from species mixing, complicating its interpretation as a temperature record. Now, the authors use RIK37 cut-off values to exclude samples that are dominated by Group II haptophytes. As calibrations exist for this phylotype, a significant portion of their data points is excluded as a consequences of this rather crude solution. I would recommend the authors to calculate the RIK38E index to better differentiate phylogeny (mixing) and derive temperatures from Group II data.

*Response:* Developing a temperature calibration using the $RIK_{38E}$ index would not resolve the issue of species mixing as both Group I and II Isochrysidales produce $C_{38}Et$ alkenones (see Zheng et al., 2019). Further temperature calibrations for Group II vary for planktonic and benthic species (see D'Andrea et al., 2016). We decided to rely on $C_{37}Me$ alkenones due to the low concentrations of $C_{38}Et$ and $C_{38}Me$ alkenones. Further, the $U_{37}^{K}$ index was successfully applied to reconstruct temperature changes in other lakes containing Group I Isochrysidales (D'Andrea et al., 2011, 2012; van der Bilt et al., 2019; Harning et al., 2020; Longo et al., 2020).

*Reviewer comment:* I commend the authors for their efforts to better constrain the seasonality of haptophyte production (and temperature sensitivity), but have two concerns. First, ice-off dictates the timing of haptophyte blooms: with this in mind, I wonder why the authors did not rely on satellite data to validate the 30 yr control run outlined in section 2.4. High-res imagery is freely available for the entire period: if in agreement with model output, this would significantly strengthen the robustness of their approach. Secondly, the presented modelling results reveal that both late winter as well as spring season temperatures help determine ice-off dates: this does not justify presenting the record as a "cod season" reconstruction.

*Response:* Thank you for the suggestion, however, the purpose of performing the sensitivity studies with the lake model is to identify the main drivers that influence lake water temperatures during the spring season and, therefore, our proxy. Validating the exact date of spring ice-off would not alter the main conclusions of our study. Further, previous work has shown that the primary alkenone bloom likely begins prior to the exact ice-off date (Longo et al., 2018). We will clarify the points discussed above in the text of section 2.4.
        We will change "cold season" to "winter and spring"/ "winter-spring" in our manuscript, where winter-spring is defined as December to May.

We have added the following sentences to section 2.4:

"Alkenone production starts prior to ice-off, then increases as the lake undergoes isothermal mixing, and decreases when thermal stratification begins to develop in late spring/early summer (Longo et al., 2018)." (lines 144-146)

"The purpose of the lake model was to determine the sensitivity of our proxy to different forcing mechanisms by assessing the magnitude of the temperature response and timing of ice-melt relative to our control simulation." (lines 150-152)

We changed "cold season" to "winter and spring"/ "winter-spring" in the manuscript.

***Reviewer comment:*** See line 215: I think the wording is far too strong here. The authors argue existing calibrations provide "unreasonable" estimates and back this up with unrealistically high temperature values. They do, however, not state that these values were calculated using site-specific intercepts provided for each of the used calibrations while the authors of the applied calibrations advise against doing so. To remedy this, I advise the authors to discuss the relative temperature fluctuations plotted in Fig. A2(b): indeed, the magnitude of these swings are of equal magnitude as those observed during the spring transitional season (Fig. 5b).

***Response:*** Thank you for the suggestion, we will modify section 3.3 to discuss the relative temperature fluctuations in Fig. A2(b).

Relative temperature changes determined using only the slope of the calibration for Group I still provide unrealistic temperature changes (for $U_{37}^K = 0.0219T$ the temperature range is 26.9°C). The slopes determined by D'Andrea et al. (2016) for Group III ($U_{37}^K = 0.0447T$) alkenone calibrations result in a smaller temperature range of 13.2°C and an estimated temperature change of 8°C from 250-350 CE to 1850-1950 CE.

We have modified section 3.3 as follows:

"The $U_{37}^K$ index can provide temperature estimates using linear relationships that are calibrated in lakes with Group I alkenone-producers (D'Andrea et al. 2011, 2016; Longo et al., 2016, 2018). Existing temperature calibrations, except for the Northern Hemisphere calibration by Longo et al. (2018), for Group I are site-specific and therefore cannot be readily applied to VGHV (e.g. calibrations give estimates of 10.2 to 33.5 °C (D'Andrea et al., 2011), 7.1 to 34.4 °C (Longo et al., 2016), 4.4 to 24.5 °C (D'Andrea et al., 2016), -1.4 to 18.3 °C (calibration for Northern Hemisphere lakes by Longo et al., 2018; see Fig. A2). Most of the variation between sites is accounted for by the y-intercept of the calibration, so the slope of Group I calibrations was suggested as a better determinant of relative temperature changes for sites lacking a site-specific calibration (D'Andrea et al., 2016). However, the slopes determined for Group I calibrations still result in a very large and likely unreasonable temperature range of 26.9 °C for $U_{37}^K = 0.0219T$ (D'Andrea et al., 2016). The slope determined for Group III alkenone calibrations ($U_{37}^K = 0.0447T$; D'Andrea et al., 2016) provides a more reasonable temperature range of 13.2 °C and an estimated temperature change of 8 °C from 250-350 CE to 1850-1950 CE. Given the sensitivity of VGHV lake water temperatures to winter and spring season perturbations and the large variability in winter and spring temperatures observed in the instrumental data (mean temperatures in the winter and spring (DJFMAM) season range from -2.4 °C to 3.4 °C with a seasonal variance of 13.1 °C between 1958-2004 at Hella station; Icelandic Meteorological Office), it is plausible to observe temperature swings close to 10 °C during the spring transitional season

(Fig. 5b). However, the amplitude of reconstructed temperatures is still relatively large considering that each sample is an average of 5-19 years, and most likely stems from the lack of a local calibration. Nevertheless, the $U^K_{37}$ index is known to be highly sensitive to temperatures in NH lakes, therefore we use the $U^K_{37}$ index to infer and evaluate qualitative changes in temperature trends and variability during the past 2,000 years." (lines 231-249)

*Reviewer comment:* Paragraph around line 230: here the authors try to relate their reconstructions to warming/cooling periods that are often referenced in the (North Atlantic) literature. I would stay clear from this and consider removing this section for a number of reasons. First, a string of recent studies has underlined just how spatio-temporally heterogeneous expression of these events is (see e.g. Werner et al. 2018 – COP, McKay et al. 2018 –GRL, and van der Bilt et al. 2019 – QSR). Secondly, most of these events are most clearly expressed in summer, while the authors argue that their record captures "cold season" conditions. Finally, and related to this, the perceived correspondence is tenuous at best as the authors also confirm by using wording like "roughly coincides" or "could be associated".

*Response:* Thank you for the suggestion, we will modify the text in section 3.3 to only discuss changes in our record. However, part of our goal is to compare our record with existing warm season reconstructions. Although there is considerable spatio-temporal variability in major warm and cold events, it is still useful to highlight anomalous time periods defined by previous warm season reconstructions in Iceland and other regions in the Northern Hemisphere. Therefore, we think it is important to keep the comparisons with warm season reconstructions in section 4.2.

We modified the text in section 3.3 as follows:

"The $U^K_{37}$ record from VGHV, corrected for species mixing, exhibits a long-term trend towards warmer spring lake water temperatures over the last 2,000 years as well as strong multi-decadal to centennial variability (Fig. 6). The gradual warming trend in our record begins after c. 400 CE. In particular, a warmer period occurs from the start of our record to c. 200 CE, followed by cooling from c. 250-600 CE. Temperature variability increases after c. 850 CE, and warmer periods occur between c. 850-1050 CE, c. 1100-1300 CE, and c. 1450-1550 CE. Relatively cooler periods occur at c. 1100-1200 CE, c. 1300-1450 CE, c. 1550-1750 CE, and c. 1850-1880 CE. However, caution should be used when interpreting results after c. 1400 CE because of low sampling resolution (c. 50 yrs between each sample)." (lines 251-257)

*Reviewer comment*: Section 4.1: please restructure and tighten this paragraph. As the presented record only covers the past 2millennia, I think the current full Holocene focus is not the right way to frame things. Also, the authors allude to the so-called "Holocene temperature Conundrum" but don's state so (or explain it clearly). The way I see things, the main message here is that spring temperatures are (not entirely surprisingly) not driven by changes in summer insolation. I would contextualize/strengthen this by discussing other non-summer temperature reconstructions (which the authors already do to some extent), and argue why one would expect to see this "cold season" imprint in a maritime Arctic setting like Iceland, where it is known that many feedbacks may overprint any radiative signature, notably surface ocean currents, but also sea-ice feedbacks – in this respect, I recommend the authors to check Park et al. 2019 – Science Advances.

*Response:* Thank you for the suggestion, we will modify the text to explain the Holocene temperature conundrum and discuss the forcings that are relevant for the last 2,000 years. As mentioned by the reviewer, radiative forcings may be overprinted by sea-ice feedbacks and circulation changes, however as we will fully discuss in the modified manuscript, this is not necessarily the case.

With regards to sea-ice feedbacks (Park et al., 2019), sea ice normally only occurs off the coast of northern Iceland, and therefore has a stronger influence on climate in northern Iceland relative to southern Iceland (Ogilvie, 1984; Ogilvie & Jónsson, 2001; Hanna et al., 2004). This observation is consistent with results from the study on mid-Holocene temperature changes in response to sea ice loss by Park et al. (2019): note the significantly lower SST response to Arctic sea ice loss (Fig. 3d) along the southern coast of Iceland (0.2 K) relative to northern Iceland (0.8 K) in the results from the mid-Holocene simulation. We will modify section 4.1 to discuss these points in more detail.

As mentioned by the reviewer, the close proximity of VGHV to the coast means that air temperatures at VGHV are also influenced by changes in sea surface temperatures (SSTs; Hanna et al., 2006). In particular, the Irminger Current, a branch of the northward moving warm waters of the North Atlantic, is advected clockwise along the southern and western coast of Iceland (e.g. Daniault et al., 2016). However, reconstructions of subpolar North Atlantic Current SSTs show diverging trends over the last 2,000 years with varying degrees of centennial to millennial variability, most likely reflecting differences in proxy seasonality (see Moffa-Sánchez et al., 2019). This makes it difficult to assess how SSTs have contributed to changes in winter and spring temperatures at VGHV, but as rightly stated by the reviewer, should not be ruled out. We will modify discussion section 4.1 to highlight the points we just discussed.

We modified section 4.1 as follows:

"Mean annual temperature syntheses from the NH exhibit a long-term cooling trend over the last 2,000 years (Kaufman et al., 2009; PAGES 2K Consortium, 2013, 2019) that is often interpreted as a response to decreasing summer insolation (Kaufman et al., 2009) and/or increased volcanic activity during the LIA (Miller et al., 2012). Climate model simulations suggest that solar variability acts as a secondary source of variability and land use changes may be important for explaining some of the changes in NH surface temperatures between the MCA and LIA, whereas increases in greenhouse gases remain stable until the late 19th century (Otto-Bliesner et al., 2016). The magnitude of the cooling trend and centennial and multi-decadal changes differs among global temperature reconstructions (PAGES 2K Consortium, 2019) and is often larger in NH temperature reconstructions compared to climate model simulations (Rehfeld et al., 2016; Ljungqvist et al., 2019). The discrepancies in temperature reconstructions and climate models could stem from a warm season bias in NH proxy reconstructions, leading to an overestimation of changes in mean annual and cold season temperatures in proxy reconstructions compared to climate model simulations (Liu et al., 2014; Rehfeld et al., 2016; PAGES 2K Consortium, 2019)." (lines 260-270)

"Iceland has a maritime climate and also sits near the edge of the Arctic sea ice; therefore, air temperatures are sensitive to regional sea-ice feedbacks and variations in sea surface temperatures (SSTs). The VGHV temperature record shows that winter and spring air temperatures warmed over the last millennium, whereas temperature and sea ice reconstructions suggest that summer air temperatures cooled in Northern and Western Iceland as sea ice increased with the coldest period occurring during the 18th and 19th centuries

(Ogilvie and Jónsson, 2001; Moros et al., 2006; Massé et al., 2008; Gathorne-Hardy et al., 2009; Axford et al., 2009, 2011; Langdon et al., 2011; Cabedo-Sanz et al., 2016; Holmes et al., 2016). Paleo- and historical records, however, indicate that sea ice was only present along the southern and western coasts of Iceland, where our study site VGHV is located, during severe ice years when sea ice is advected clockwise around the country (Ogilvie, 1996; Axford et al., 2011; Cabedo-Sanz et al., 2016). Similarly, millennial-scale changes in spring temperatures inferred from biogenic silica in western Iceland are decoupled from temperature and sea-ice changes in Northern Iceland, suggesting that spring temperatures are likely more sensitive to changes in regional SSTs (Geirsdóttir et al., 2009). SST reconstructions near southern Iceland show that surface temperatures either increased (Berner et al., 2008; Thornalley et al., 2009; Miettinen et al., 2012; Orme et al., 2018) or did not significantly change (Sicre et al., 2011; Van Nieuwenhove et al., 2018) over the last 2,000 years. Marine reconstructions of temperature from below the summer thermocline and bottom water record a decrease in mean annual temperatures over the Common Era (Thornalley et al., 2009; Ólafsdóttir et al., 2010; Moffa-Sánchez et al., 2014) as the transport of warm North Atlantic Current waters by the Irminger Current decreased over the last 2,000 years (Ólafsdóttir et al., 2010). Based on existing paleo- and historical records we conclude that sea ice feedbacks only play a minor role in driving long-term changes in winter and spring temperatures at our study site, whereas an increase in SSTs along the southern coast could contribute to the warming trend observed in our record. However, discrepancies in existing proxy records makes it difficult to correlate changes in SSTs to changes in winter and spring temperatures at VGHV." (lines 281-301)

*Reviewer comment:* Finally, as the authors point out in section 2.4, Iceland receives little sunlight during winter: I therefore recommend them to plot early spring insolation in Fig. 6a instead of winter + spring insolation.

*Response:* We will modify Fig. 6a to plot spring and winter insolation separately.

We modified Fig. 6a to plot spring and winter insolation separately.

*Reviewer comment:* Section 4.2: the authors (partly) attribute higher-frequency changes to shifts in regional climate dynamics, notably the NAO. When doing so, it would be most helpful to provide a contextual understanding of this complex system on Iceland – what happens to the different components of the regional climate system during (shifts between) positive/ negative NAO phases. Now, it oft feels as if this discussion is shoehorned into an NAO mould using a hotchpotch of sources. Also, respect the sampling and chronological resolution of this dataset: I don't think it warrants attribution to multi-annual forcing mechanisms.

*Response:* Thank you for the suggestion. We will update the text in section 4.2 to discuss forcings that are important on multi-decadal timescales and how they influence the regional climate of Iceland, particularly during the winter and spring season.
    As discussed below and as we will explain in the modified text, it is hypothesized that low frequency changes in instrumental and paleoclimate archives from the North Atlantic region are driven by variability in the NAO (e.g. Hurrell, 1995; Pinto & Raible, 2012; Ortega et al., 2015). A recent study demonstrated that NAO variability on interannual to decadal timescales is most likely dominated by meridional shifts in the jet stream and storms tracks, whereas on

multi-decadal timescales NAO variability is associated with changes in the speed and strength of the storm tracks (Woolings et al., 2015). On multi-decadal timescales studies have also linked variability in the NAO to changes in sea ice (Delworth et al., 2016), the Atlantic meridional overturning circulation (Delworth et al., 2016), and the Atlantic Multi-decadal Variability (Omrani et al., 2014, 2016; Peings & Magnusdottir, 2014).

We modified the text in section 4.2 as follows:

"The NAO, defined by differences in sea-level pressure between the subpolar low and the subtropical high, is a major source of atmospheric variability during the winter months (Hurrell, 1995). Although the NAO is mainly associated with interannual timescales, there is also evidence that NAO-like patterns can emerge at multi-annual to centennial timescales, potentially linked to coupled changes in oceanic and atmospheric circulation (Visbeck et al., 2003; Delworth et al., 2016; Yeager and Robson 2017). For instance, a positive (negative) NAO phase that persists for multiple winters can lead to increased (decreased) deepwater formation in the Labrador Sea and strengthening (weakening) of the subpolar gyre and the meridional overturning circulation, thereby resulting in an increased (decreased) transport of warm waters towards the poles (Eden and Jung, 2001; Visbeck et al., 2003; Latif et al., 2006). Alternatively, an increase in the southward transport of polar waters could also result in a reduction of deepwater formation in the Labrador Sea and a weaker subpolar gyre, leading to a decrease in northward oceanic heat transport and centennial cooling of ocean and regional air temperatures (Moffa-Sánchez and Hall, 2017; Moreno-Chamarro et al., 2017)." (lines 348-358)

"In contrast, terrestrial records from the Arctic and Northern Europe indicate that temperatures were on average cooler between c. 450-700 CE and c. 500-650 CE, respectively (Kaufman et al., 2009; Sigl et al., 2015; Helama et al., 2017). A peak in sea ice is also recorded in a high-resolution $IP_{25}$ reconstruction from the North Icelandic shelf (Cabedo-Sanz et al., 2016), whereas lower resolution records that were developed using quartz and $IP_{25}$ show a gradual increase in sea ice after c. 400 CE but not distinct peak (Moros et al., 2006; Cabedo-Sanz et al., 2016). Increases in sea ice during this time period are attributed to a southward shift of the subpolar front and the increased advection of drift ice from Greenland to Northern Iceland (Moros et al., 2006; Cabedo-Sanz et al., 2016), leading to cooler winter and spring temperatures in Northern and Southern Iceland." (lines 369-376)

"Summer temperature reconstructions from lakes in Northern and Western Iceland also record warmer temperatures c. 800-1300 CE but with distinct cold excursions occurring between c. 1000-1300 CE (Axford et al., 2009; Gathorne-Hardy et al., 2009; Holmes et al., 2016). Peaks in sea ice, however, are only noted after c. 1200 CE (Ogilvie, 1992; Ogilvie and Jónsson, 2001; Massé et al., 2008; Cabedo-Sanz et al., 2016), suggesting that an alternative mechanism, such as the NAO, may be responsible for the short-term variability observed in terrestrial temperature records from Iceland during this time period." (lines 383-387)

"In the VGHV record, multi-decadal variability and an inconsistent temperature response to major radiative forcings during the LIA suggest that temperature anomalies during the winter and spring are driven by both forced and unforced variability. For instance, the strong negative radiative forcing after the Samalas eruption (1258 CE) and the Wolf solar minimum (c. 1280-1350 CE) correspond to an increase in drift ice along the North Icelandic shelf (Fig.

7; Massé et al., 2008; Cabedo-Sanz et al., 2016), a cold excursion in winter subsurface temperatures from the North Icelandic shelf c. 1350-1500 CE (Harning et al 2019), and cooling in the VGHV temperature record. Similarly, the cumulative effects of the Dalton solar minimum c. 1790-1830 CE and multiple major volcanic eruptions (i.e., Laki 1783 CE, unidentified 1809 CE, and Tambora 1815 CE; Sigl et al., 2015; Toohey and Sigl, 2017) in the late 18th and early 19th century could have resulted in enhanced sea ice feedbacks and cooling in VGHV temperatures between c. 1800-1900 CE (Massé et al., 2008; Zanchettin et al., 2012; Cabedo-Sanz et al., 2016). The inconsistent response in VGHV temperatures to volcanic eruptions and solar minima between c. 1450-1750 CE could be associated with stochastic climate processes, such as the NAO, that counteracted the effects of negative radiative forcings and led to winter warming over Iceland rather than cooling.

On multi-decadal to centennial timescales, changes in the VGHV record do not consistently correspond to major temperature anomalies observed during the summer months. The differences in seasonal climate responses to external forcings imply that the regional manifestation of these events depends on the initial state of the atmosphere and ocean but is also modulated by internal climate variability (Zanchettin et al., 2012; Otto-Bliesner et al., 2016; Anchukaitis et al., 2019)." (lines 409-425)

**References**

D'Andrea, W. J., Huang, Y., Fritz, S. C., and Anderson, N. J.: Abrupt Holocene climate change as an important factor for human migration in West Greenland. Proc. Natl. Acad. Sci., 108, 9765-9769, https://doi.org/10.1073/pnas.1101708108, 2011.

D'Andrea, W. J., Vaillencourt, D. A., Balascio, N. L., Werner, A., Roof, S. R., Retelle, M., and Bradley, R. S.: Mild Little Ice Age and unprecedented recent warmth in an 1800 year lake sediment record from Svalbard. Geology, 40, 1007-1010, https://doi.org/10.1130/G33365.1, 2012.

D'Andrea, W. J., Theroux, S., Bradley, R. S., and Huang, X.: Does phylogeny control $U_{37}^K$ - temperature sensitivity? Implications for lacustrine alkenone paleothermometry. Geochim. Cosmochim. Acta, 175, 168-180, https://doi.org/10.1016/j.gca.2015.10.031, 2016.

Daniault, N., Mercier, H., Lherminier, P., Sarafanov, A., Falina, A., Zunino, P., Pérez, F.F., Ríos, A.F., Ferron, B., Huck, T. and Thierry, V.: The northern North Atlantic Ocean mean circulation in the early 21st century. Prog. Oceanogr., 146, 142-158, 2016.

Delworth, T. L., Zeng, F., Vecchi, G. A., Yang, X., Zhang, L., and Zhang, R.: The North Atlantic Oscillation as a driver of rapid climate change in the Northern Hemisphere. Nat. Geosci., 9, 509-512, 2016.

Hanna, E., Jónsson, T., Ólafsson, J., and Valdimarsson, H.: Icelandic coastal sea surface temperature records constructed: putting the pulse on air–sea–climate interactions in the northern North Atlantic. Part I: comparison with HadISST1 open-ocean surface temperatures and preliminary analysis of long-term patterns and anomalies of SSTs around Iceland. J. Clim., 19, 5652-5666, https://doi.org/10.1175/JCLI3933.1, 2006.

Harning, D. J., Curtin, L., Geirsdóttir, Á., D'Andrea, W. J., Miller, G. H., and Sepúlveda, J.: Lipid biomarkers quantify Holocene summer temperature and ice cap sensitivity in Icelandic lakes. Geophys. Res. Lett., 47, 2020.

Hurrell, J.W.: Decadal trends in the North Atlantic Oscillation: regional temperatures and precipitation. Science, 269, 676-679, https://doi.org/10.1126/science.269.5224.676, 1995.

Longo, W.M., Huang, Y., Yao, Y., Zhao, J., Giblin, A.E., Wang, X., Zech, R., Haberzettl, T., Jardillier, L., Toney, J., and Liu, Z.: Widespread occurrence of distinct alkenones from Group I haptophytes in freshwater lakes: Implications for paleotemperature and paleoenvironmental reconstructions. Earth Planet. Sci. Lett., 492, 239-250, https://doi.org/10.1016/j.epsl.2018.04.002, 2018.

Longo, W. M., Huang, Y., Russell, J. M., Morrill, C., Daniels, W. C., Giblin, A. E., and Crowther, J.: Insolation and greenhouse gases drove Holocene winter and spring warming in Arctic Alaska. Quat. Sci. Rev., 242, 2020.

Hanna, E., Jónsson, T., and Box, J. E.: An analysis of Icelandic climate since the nineteenth century. Int. J. Climatol.: J. Roy. Meteor. Soc., 24, 1193-1210, https://doi.org/10.1002/joc.1051, 2004.

Hurrell, J. W., and Deser, C.: North Atlantic climate variability: the role of the North Atlantic Oscillation. J. Mar. Sys., 79, 231-244, 2010.

Moffa-Sánchez, P., Moreno-Chamarro, E., Reynolds, D.J., Ortega, P., Cunningham, L., Swingedouw, D., Amrhein, D.E., Halfar, J., Jonkers, L., Jungclaus, J.H. and Perner, K.: Variability in the northern North Atlantic and Arctic oceans across the last two millennia: A review. Paleoceanogr. Paleoclimatol., 34, 1399-1436, 2019.

Ogilvie, A. E.: The past climate and sea-ice record from Iceland, Part 1: Data to AD 1780. Clim. Change, 6, 131-152, 1984.

Ogilvie, A. E., and Jónsson, T.: " Little ice age" research: A perspective from Iceland. Clim. Change, 48, 9-52, 2001.

Ólafsdóttir, K. B., Geirsdóttir, Á., Miller, G. H., and Larsen, D. J.: Evolution of NAO and AMO strength and cyclicity derived from a 3-ka varve-thickness record from Iceland. Quat. Sci. Rev., 69, 142-154, 2013.

Omrani, N. E., Keenlyside, N. S., Bader, J., and Manzini, E.: Stratosphere key for wintertime atmospheric response to warm Atlantic decadal conditions. Clim. Dyn., 42, 649-663, https://doi.org/10.1007/s00382-013-1860-3, 2014.

Omrani, N. E., Bader, J., Keenlyside, N. S., and Manzini, E.: Troposphere–stratosphere response to large-scale North Atlantic Ocean variability in an atmosphere/ocean coupled model. Clim. Dyn., 46, 1397-1415, https://doi.org/10.1007/s00382-015-2654-6, 2016.

Ortega, P., Lehner, F., Swingedouw, D., Masson-Delmotte, V., Raible, C. C., Casado, M., and Yiou, P.: A model-tested North Atlantic Oscillation reconstruction for the past millennium. Nature, 523, 71-74, 2015.

Park, H. S., Kim, S. J., Stewart, A. L., Son, S. W., and Seo, K. H.: Mid-Holocene Northern Hemisphere warming driven by Arctic amplification. Sci. Adv., 5, 2019.

Peings, Y., and Magnusdottir, G.: Wintertime atmospheric response to Atlantic multidecadal variability: Effect of stratospheric representation and ocean–atmosphere coupling. Clim. Dyn., 47, 1029-1047, https://doi.org/10.1007/s00382-015-2887-4, 2016.

Pinto, J. G., & Raible, C. C.: Past and recent changes in the North Atlantic oscillation. Wiley Interdiscip. Rev. Clim, 3, 79-90, 2012.

van der Bilt, W. G., D'Andrea, W. J., Werner, J. P., and Bakke, J.: Early Holocene temperature oscillations exceed amplitude of observed and projected warming in Svalbard lakes. Geophys. Res. Lett., 46, 14732-14741, 2019.

Woollings, T., Franzke, C., Hodson, D. L. R., Dong, B., Barnes, E. A., Raible, C. C., and Pinto, J. G.: Contrasting interannual and multidecadal NAO variability. Clim. Dyn., 45, 539-556, 2015.

Zheng, Y., Heng, P., Conte, M. H., Vachula, R. S., and Huang, Y.: Systematic chemotaxonomic profiling and novel paleotemperature indices based on alkenones and alkenoates: Potential for disentangling mixed species input. Org. Geochem., 128, 26-41, 2019.

**Response to Anonymous Referee #2**

We would like to thank the referee for taking the time to review our manuscript and for their comments. We have included the reviewer comments below along with our responses and edits to the manuscript (highlighted in blue).

*Reviewer comment:* Richter et al. reconstruct cold-season temperature trends over the last 2 ky for south- west Iceland using alkenones produced by lacustrine haptophyte algae. The authors demonstrate that alkenones from lake VGHV record a long-term warming trend, as well as decadal to centennial scale variability within the long-term trend. They couple this temperature reconstruction with a lake energy balance model to support that increasing high-latitude winter insolation is likely responsible for the overarching cold-season warming for the last ~2 ky, while climate perturbations are likely responsible for high frequency variability in proxy data. The authors contextualize this data in a broader framework by suggesting that this dataset, and more studies like it, could help consolidate discrepancies between global climate model output and proxy reconstructions for the northern hemisphere through the Holocene. Major contributions of this manuscript:

This work offers important insight into seasonal differences in temperature for SW Iceland (and by inference this part of the N Atlantic) for the last 2 ky

The coupling of proxy inference and lake energy model is a progressive approach for interpreting proxy data by testing it within varying climate forcing scenarios

The presentation of the data is thoughtful and clear

Criticisms of the manuscript:

I find the discussion around the seasonality of this proxy, and the conclusions drawn from it, to be somewhat confusing and at times inconsistent. I think it would be helpful if the manuscript more clearly articulated the chain of logic/evidence that provides that alkenones, which are stated to bloom in spring, can be interpreted more broadly as a record of cold-season temperatures driven by cold-season insolation.

Within this point, I would find it helpful if the background discussion around the proxy touched on the fidelity of alkenones for reconstructing temperature (is it known to have significant error associated with it, or low significance values?) and are there calibration data that covers a climatically similar region? I appreciate that this record is being interpreted qualitatively and it is clearly stated in the manuscript that there is no local calibration data, but I think it would improve confidence in this interpretation of the data to know that it has been tested/utilized in comparable locations, particularly in interpreting high frequency changes as related to climate perturbations and not stochastic proxy noise

*Response:* We will update the manuscript to include a discussion of previous Group I alkenone calibrations, their fidelity, and previous downcore records as discussed below.

   Group I Isochrysidales and their corresponding alkenones have, so far, only been identified in Northern Hemisphere lakes at latitudes ranging from 42-81°N (Longo et al., 2018). The Northern Hemisphere lake calibration for Group I alkenones, which includes VGHV and was developed using the average temperature of the four months centered around the spring isotherm for each lake ($U_{37}^{K} = 0.029T-0.49$, $r^2 = 0.60$), has an RMSE $= \pm 1.69$°C

(Longo et al., 2018). An updated calibration for Group I that includes additional lakes in northeastern China ($U_{37}^K = 0.030T\text{-}0.479$, $r^2 = 0.0479$) has an RMSE = $\pm$ 1.71°C (Yao et al., 2019). Group I alkenone calibrations also exist for Lake BrayaSø in Greenland ($U_{37}^K = 0.0245T\text{-}0.779$, $r^2 = 0.96$, note the calibration also includes data from several German lakes, see Zink et al., 2001; D'Andrea et al., 2011), Lake Kongressvatnet in Svalbard ($U_{37}^K = 0.0255T\text{-}0.804$, $r^2 = 0.85$, D'Andrea et al., 2012), Toolik Lake in Alaska ($U_{37}^K = 0.021T\text{-}0.68$, $r^2 = 0.85$; Longo et al., 2016), and Vikvatnet in Norway ($U_{37}^K = 0.0284T\text{-}0.655$, $r^2 = 0.94$; D'Andrea et al., 2016). A key argument linking our data to winter conditions is that the haptophyte bloom time may be fixed by the annual cycle by processes such as the photoperiod, such that blooms may develop in the very early stages of ice-off. As discussed in the paper, the timing of ice-off is set in part by winter conditions and partly by early spring temperatures (which we refer to in the text as "winter-spring"). Thus, although the haptophyte bloom occurs in spring, spring lake temperatures are set in part by winter temperatures. Our modeling work supports the seasonal dependence of spring temperatures at our study site. We will elaborate on these points in the text and we will modify Figure A2 to include the standard error for each calibration that is plotted.

There are only a few studies that have applied Group I alkenones to downcore records. A 16,000-year reconstruction of winter-spring (DJFMAM) temperatures was developed using Group I alkenones for Lake E5 in Northern Alaska, and also exhibits gradual warming throughout the middle to late Holocene in response to increasing winter-spring insolation, greenhouse gases, and regional feedbacks (Longo et al., 2020). Temperature reconstructions with a comparable resolution to our record include reconstructions from Lake BrayaSø in Greenland that spans c. 5,600 yrs BP (resolution c. 7-90 yrs, D'Andrea et al., 2011) and Kongressvatnet in Svalbard that spans 1,800 yrs BP (resolution 4-30 yrs, D'Andrea et al., 2012). In both studies, the Group I alkenone records are interpreted as summer (JJA) temperature reconstructions due to the very late ice-off dates in these regions (D'Andrea et al., 2011, 2012). The amplitudes of the temperature changes observed in the temperature records from Greenland (temperatures range from 3°C to 9°C to between 10 CE and 1999 CE; D'Andrea et al., 2011) and Svalbard (temperatures range from 2°C to 6°C between 230 CE and 2009 CE; D'Andrea et al., 2012) are smaller in magnitude than the estimated temperature change in our record using $U_{37}^K = 0.029T$ (temperature range 20°C). The higher amplitudes observed in our reconstruction could be explained by the lack of a local calibration and that our record reflects variations in winter and spring temperatures rather than summer temperatures. We will modify our discussion to highlight the studies we just discussed.

We have modified section 2.3 to discuss previous temperature calibrations and reconstructions using alkenones from lake sediments as follows:

"Group I Isochrysidales and their corresponding alkenones have, so far, only been identified in Northern Hemisphere lakes at latitudes ranging from 42-81 °N (Longo et al., 2018; Richter et al., 2019). The Northern Hemisphere lake calibration for Group I alkenones, which includes VGHV, was developed using the average spring temperatures for each lake during ice-off and the main Group I Isochrysidales bloom ($U_{37}^K = 0.029T\text{-}0.49$, $r^2 = 0.60$, RMSE = $\pm$ 1.69°C; Longo et al., 2018). An updated calibration for Group I that includes additional lakes in northeastern China ($U_{37}^K = 0.030T\text{-}0.479$, $r^2 = 0.0479$) has an RMSE = $\pm$ 1.71°C (Yao et al., 2019). Group I alkenone calibrations also exist for Lake BrayaSø in Greenland ($U_{37}^K = 0.0245T\text{-}0.779$, $r^2 = 0.96$, note the calibration also includes data from several German lakes, see Zink et al., 2001; D'Andrea et al., 2011), Lake Kongressvatnet in Svalbard ($U_{37}^K =$

0.0255$T$-0.804, $r^2 = 0.85$, D'Andrea et al., 2012), Toolik Lake in Alaska ($U_{37}^K = 0.021T$-0.68, $r^2 = 0.85$; Longo et al., 2016), and Vikvatnet in Norway ($U_{37}^K = 0.0284T$-0.655, $r^2 = 0.94$; D'Andrea et al., 2016). Temperature calibrations using the $U_{37}^K$ index were successfully applied to develop high resolution records of summer temperatures in Greenland (c. 5,600 yrs BP; D'Andrea et al., 2011) and Svalbard (1,800 yrs BP; D'Andrea et al., 2012) and a winter-spring temperature record in Alaska (16,000 yrs BP; Longo et al., 2020). However, regional variability in the relationship between the $U_{37}^K$ index and temperature requires the development of local temperature calibrations (Wang and Liu, 2013; D'Andrea et al., 2016; Longo et al., 2016). Unfortunately, there is currently no local calibration for Icelandic lakes." (lines 128-141)

In addition, we have modified the discussion to include previous alkenone reconstructions of winter and spring temperatures:

"For instance, pollen records of cold-season temperatures from North America and Europe (Mauri et al., 2015; Marsicek et al., 2018) and an alkenone reconstruction of winter-spring temperatures from Alaska (Longo et al., 2020) suggest that increasing winter and spring orbital insolation over the Holocene drove warming during the winter and spring season." (lines 304-307)

***Reviewer comment:*** I would find it valuable to know if there is a competing effect from declining summer insolation/temperature on spring temperatures and the timing of ice-out. It's unclear if JJA/SON is held constant in the model, or if there is little response to ice-out date/water temps given changing temps/insolation during these seasons (Fig. 5).

***Response:*** As described in section 2.4, perturbations in insolation and temperature are applied to every season (DJF, MAM, JJA, and SON) to determine the effects of these seasonal perturbations on spring lake water temperatures and ice-out dates. In section 3.2 we find that the there is no competing effect of summer or fall insolation and air temperature on spring lake water temperatures and the timing of ice-out. More generally, winter temperatures at VGHV are always cold enough to freeze the lake surface, and because the minimum water temperature is always reached during the winter by this process, the summer climate has a small influence relative to early spring temperatures. Thereby, the seasonal lacustrine cycle effectively 'resets' the temperature each winter. We will modify sections 3.2 and 4.1 to make these points clearer.

We have added the following sentences to clarify the points discussed above:

"There are no competing effects of summer (JJA) or fall (SON) insolation and air temperature on spring lake water temperatures and the timing of ice-out." (Section 3.2, lines 217-219)

"Lake water temperatures in VGHV solely respond to changes during the winter and spring season because the lake re-freezes every winter and reaches minimum lake water temperatures, meaning any influence from the previous summer or fall season are negligible (e.g. Assel and Robertson, 1995)." (Section 4.1, lines 274-276)

***Reviewer comment:*** Is the lake model consistent with observational data for what controls lake ice-out dates & water temps? (i.e. are there examples in modern observational data of earlier ice-off dates in regions with increasing winter air temperatures?)

*Response:* In a previous study, controls on lake ice-out dates and water temperatures were determined for Toolik Lake in Alaska and were validated using available monitoring data demonstrating that increasing winter air temperatures led to earlier ice-off dates (see Longo et al., 2020). However, it would be extremely challenging given the existing climatological data to validate the model at our study site. We do not attempt to quantitatively interpret the temperature data, so it is not entirely necessary for our study to perform this validation; our modeling work is focused on sensitivity tests.

We have added the following sentence to section 2.4 to clarify the purpose of our lake model:

"The purpose of the lake model was to determine the sensitivity of our proxy to different forcing mechanisms by assessing the magnitude of the temperature response and timing of ice-melt relative to our control simulation." (section 2.4, lines 150-152)

*Reviewer comment:* Some discussion around if the parameters & outputs of this model are climatically probable for the coverage of the record would improve this manuscript. E.g. An air temperature increase in the winter of +7 deg does dramatically move the ice-out date, but that change in temperature seems far too large for the amount of cold-season insolation change. These bounds, I think, are justified in lines 145-157, but I find this statement/constraint confusing. Observed temperatures in any given season can range by +/- 7, but why would average cold-season temperatures range by this amount over the last 2 ky? Change in insolation seems to have much less impact in ice-out dates in the model (Fig 5) but is credited for driving trends in alkenone data. This reads as a mismatch in results-conclusions as written, and the manuscript would improve from some text that consolidates the results of the model with their interpretation of the data.

*Response:* We will modify section 2.4 to elaborate on the temperature bounds used in the lake model as discussed below.

The temperature perturbations of the lake model are used to determine the sensitivity of lake water temperatures and ice-off dates to seasonal changes in temperature, and thereby confirm the seasonality of our proxy. The magnitude of the temperature perturbations used in the lake model are based on instrumental data at Hella station in Iceland (1958-2004, Icelandic Meteorological Office). Between 1958-2004 the range of mean seasonal temperatures are as follows: winter (DJF) -3.7°C to 1.8°C, spring (MAM) -1.0°C to 6.9°C, summer (JJA) 8.8°C to 12.0°C, and fall (SON) -1.3°C to 6.7°C. From year-to-year, ±7°C swings in mean seasonal temperatures, particularly during the transitional seasons, are reasonable based on the instrumental data. Further, we use these large changes in temperature to demonstrate that large fluctuations in summer and fall temperatures have a minimal influence on our proxy. We could certainly perform the sensitivity tests using a larger temperature range, but there is little constraint on where the bounds of that range might lie. We acknowledge that each of our samples represent an average temperature of 5-19 years and, based on the instrumental data, we would then expect seasonal air temperatures to change by ±3°C. However, as mentioned in our previous comment, reconstructions from Greenland and Svalbard with a similar resolution also report 4-6°C changes in temperature over the last c. 2,000 years (D'Andrea et al., 2011, 2012).

One of the goals of our study is to determine what might lead to a long-term increase in the winter-spring temperatures observed in our record. As demonstrated in our model, lake water temperatures are sensitive to both changes in air temperature and also directly respond to changes in shortwave radiation. Short-term and long-term changes in air temperatures can be attributed to multiple factors (i.e., shortwave radiation, greenhouse gases, and regional

feedbacks), which, as demonstrated in the lake model, can either amplify or overprint the changes observed in the lake water temperatures at our study site. We will clarify these points in our discussion section.

We have modified section 2.4 to highlight the points discussed above as follows:

"Between 1958-2004 the range of mean seasonal temperatures are as follows: winter (DJF) -3.7 °C to 1.8 °C, spring (MAM) -1.0 °C to 6.9 °C, summer (JJA) 8.8 °C to 12.0 °C, and fall (SON) -1.3 °C to 6.7 °C. To constrain the seasonality of our proxy, we perturbed the ERA interim seasonal air temperature values by -7 °C, -3 °C, 0 °C, +3 °C, and +7 °C and re-ran the lake model with the adjusted parameters." (section 2.4, lines 158-161)

*Reviewer comment:* The warming trend apparent in this data really seems to start just before ~400 CE, with temperature values only returning to average values from the start of the record several centuries later. This could indicate the long-term trend is less pronounced than what is captured by this window (i.e. there is a rebounding from depressed temperatures, and then warming above that average only over the last millennia). I think it would be an interesting point to add to consider if the early values (0-200 CE) are the anomaly of the record, of if the record should be considered in the context of these early values.

*Response:* Thank you for the interesting suggestion and we will include this in our discussion. To verify that the long-term trend is less pronounced than what is captured in the current window, we would need to extend our record beyond 0 CE. That will hopefully will be pursued in future studies either in VGHV or other nearby sites in Iceland that contain Group I alkenones.

We have updated discussion section 4.1 to include the following sentence:

"However, as our record only spans the last 2,000 years, the warming trend in our record could also be a feature of the last millennium associated with regional processes." (lines 279-280)

*Reviewer comment:* Overall I think this is a significant and important study, but the manuscript would benefit from some additions to background information and from adding text that consolidates what is learned in the model with their proxy data, and the climate implications of these data, so that it is very clear the conclusions are supported by their data prior to publication.

*Response:* Thank you, and we hope that the changes we outline above address this comment.

**References**

D'Andrea, W. J., Huang, Y., Fritz, S. C., and Anderson, N. J.: Abrupt Holocene climate change as an important factor for human migration in West Greenland. Proc. Natl. Acad. Sci., 108, 9765-9769, https://doi.org/10.1073/pnas.1101708108, 2011.

D'Andrea, W. J., Vaillencourt, D. A., Balascio, N. L., Werner, A., Roof, S. R., Retelle, M., and Bradley, R. S.: Mild Little Ice Age and unprecedented recent warmth in an 1800 year lake sediment record from Svalbard. Geology, 40, 1007-1010, https://doi.org/10.1130/G33365.1, 2012.

D'Andrea, W. J., Theroux, S., Bradley, R. S., and Huang, X.: Does phylogeny control $U^K_{37}$ -temperature sensitivity? Implications for lacustrine alkenone paleothermometry. Geochim. Cosmochim. Acta, 175, 168-180, https://doi.org/10.1016/j.gca.2015.10.031, 2016.

Icelandic Meteorological Office: https://en.vedur.is/climatology/data/, last access: 11 June 2020.

Longo, W. M., Theroux, S., Giblin, A. E., Zheng, Y., Dillon, J. T., and Huang, Y.: Temperature calibration and phylogenetically distinct distributions for freshwater alkenones: evidence from northern Alaskan lakes. Geochim. Cosmochim. Ac., 180, 177-196, https://doi.org/10.1016/j.gca.2016.02.019, 2016.

Longo, W.M., Huang, Y., Yao, Y., Zhao, J., Giblin, A.E., Wang, X., Zech, R., Haberzettl, T., Jardillier, L., Toney, J., and Liu, Z.: Widespread occurrence of distinct alkenones from Group I haptophytes in freshwater lakes: Implications for paleotemperature and paleoenvironmental reconstructions. Earth Planet. Sci. Lett., 492, 239-250, https://doi.org/10.1016/j.epsl.2018.04.002, 2018.

Longo, W. M., Huang, Y., Russell, J. M., Morrill, C., Daniels, W. C., Giblin, A. E., and Crowther, J.: Insolation and greenhouse gases drove Holocene winter and spring warming in Arctic Alaska. Quat. Sci. Rev., 242, 2020.

Yao, Y., Zhao, J., Longo, W.M., Li, G., Wang, X., Vachula, R.S., Wang, K.J. and Huang, Y.: New insights into environmental controls on the occurrence and abundance of Group I alkenones and their paleoclimate applications: Evidence from volcanic lakes of northeastern China. Earth Planet. Sci. Lett., 527, 2019.

Zink, K. G., Leythaeuser, D., Melkonian, M., and Schwark, L.: Temperature dependency of long-chain alkenone distributions in recent to fossil limnic sediments and in lake waters. Geochim. Cosmochim. Acta, 65, 253-265, 2001.

---

## Author Response (AR2)

***Response to reviewer comments:***

Editor Decision: Reconsider after major revisions (03 Mar 2021) by Bjørg Risebrobakken
Comments to the Author:
Dear Nora Richter et al.,

Thank for very much for submitting your revised "Winter-spring warming in the North Atlantic during the last 2,000 years: Evidence from Southwest Iceland" manuscript. I have now received two evaluations of the revisions done. One still recommends major revisions, and both raise questions that I recommend you to address carefully. In part these comments reflect issues raised by both reviewers in the first review round, but not taken into account. There are also questions raised with respect to the interpretation relative to the strength of the suggested forcing.

I would also like to ask for more background information to be provided with respect to the model setup. According to Dee et al., 2018, lake specific tuning of parameters is necessary, in addition to the user defined initial conditions. Please provide information on choices made for local tuning of the model, as well as specification of individual input variables and sources of these. And how well does your control simulation represent the known lake conditions? As far as I can see you show no validation of the model set up for your lake? In part this links back to one of the main concerns of the reviewer, related to assessment of the model.

I will ask for a revision of the manuscript, taking into account the concerns still raised. I will look forward to seeing the revised version and your responses to the reviewer's comments. Please provide responses to each comment raised and a version of the revised manuscript including track changes highlighting how the changes have been implemented.

Thank you very much for submitting your work to Climate of the Past.

Best regards,
Bjørg Risebrobakken
Editor, Climate of the Past

***Response:*** Thank you for taking the time to review our paper and for your feedback. We have included a more detailed description of the lake model parameterization and validation of our control run in the manuscript. Please see our response to reviewer 1 below for a more detailed description of what we modified. We hope you that you will consider our manuscript for publication with the corrections outlined below.

**Reviewer 1**

Dear authors,

*Reviewer:* I have now read your rebuttal to my comments and those made by another reviewer. In light of these, I have additional suggestions that ought to be reflected before publication in my opinion:

*Response:* Thank you for taking the time to review our manuscript and for your feedback

*Reviewer:* RIK38E: my previous comment concerning the use of this index may have been mis-read. I did not suggest to use RIK38E to calibrate your samples, but merely to (better) distinguish between Group I and 2 producers (see Longo et al. 2016/2018). As I said in my initial comments, separating them is quite critical for robust temperature inferences, and using RIK38E values may significantly refine this distinction.

*Response:* We agree that it is crucial to distinguish between Group I and II alkenones, and we apologize for misinterpreting your previous comment. We included an additional figure in our appendix that compares the $RIK_{37}$ and $RIK_{38E}$ values (Fig. A1). Unfortunately, the concentrations of the $C_{38:3}Et$ and its isomer were sometimes too low to reliably identify and quantify those compounds, therefore we only included the samples where we were able to reliably quantify $C_{38:3}Et$ and its isomer. As you will observe in our figure, all of the samples that we include in our manuscript fall within the range of Group I Isochrysidales as previously determined by Longo et al. (2018). We updated section 2.3 to include the $RIK_{38}$ equation and we added the following text:

"$RIK_{38E}$ values of 0 to 0.57 were empirically shown to correspond to Group I alkenone distributions, whereas Group II alkenone distributions correspond to values between 0.75 to 1 (Longo et al., 2016, 2018)." (section 2.3, lines 131-133)

We also added the following text to section 3.1:

"To further validate our results, we also did an additional comparison using both the $RIK_{37}$ and $RIK_{38E}$ indices (Fig. A1). Due to lower abundance of $C_{38:3}$ Et and its isomer we had fewer datapoints for comparison, but our results still demonstrate that datapoints used in our study are predominantly produced by Group I Isochrysidales." (section 3.1, lines 226-228)

*Reviewer:* Ice-off dates: both me and the other reviewer of this manuscript encourage the authors to assess their model by using free and easily available observational data to assess the potential impact of variable ice-off dates. Yet, no steps have been undertaken in the revised manuscript to do so, while the authors do use observational data to justify their calibration. I strongly encourage the authors to take this comment to heart, especially in light of the high-amplitude temperature shifts that they infer.

*Response:* We think it is important to understand that we are not attempting to argue that the lake model precisely constrains the timing of ice-out nor spring temperatures at our study site. Rather, we use the model to explore the sensitivity of ice-out and temperature to different forcings. We modified Fig. 5 and the outputs from our sensitivity tests to reflect temperature anomalies and changes in ice-off dates relative to our control simulation.

Nevertheless, to address these concerns we have included a table with the lake-specific parameters used to calibrate our lake model in the appendix (see Table A1). As you will see in the table, the parameters were constrained based on data in Blair et al. (2015) and a previous simulation by Longo et al. (2020) on lakes in Northern Alaska where the lake model was validated with limnological data from the Toolik Environmental Data Center and the Arctic Long-Term Ecological Research program over a 6-year period. Lake E5 in Northern Alaska is similar in size and with a similar catchment area to VGHV (Longo et al., 2020). Similar to Longo et al. (2020), we also found that the neutral drag coefficient has the largest influence on outputs from the lake model simulation. In addition, we summarized the available data in the literature and the few observations we were able to make using satellite data in Tables A2 and A3, respectively. However, our study site was often obscured by clouds during the spring season, imposing significant limits on this exercise. We recorded the last day ice was observed on the VGHV and the first day where the lake was completely ice-free. We also included a comparison of the simulated lake surface temperatures and changes in ice-cover from our control run with air temperature data from a nearby meteorological station (Hella Station), since lake temperature time-series data is not available from our study site. Based on the available data in the literature, meteorological data, and satellite imagery our lake parametrization reflects the general conditions observed in southwest Iceland. To further refine the lake model simulation for VGHV and use the proxy system model developed by Dee et al. (2018) we would need a much longer observational dataset (see Longo et al. 2020).

We have updated the text in sections 2.4 and 3.2 to reflect these changes:

"We investigated the controls on spring lake water temperatures and the timing of ice-melt in VGHV using a lake energy balance model (Dee et al., 2018). The purpose of the lake model was to determine the sensitivity of our proxy to different forcing mechanisms by assessing the temperature response and timing of ice-melt relative to our control simulation. Due to the lack of extensive observational datasets from VGHV to test our model, we adjusted the initial parameters using available data in the literature and parametrizations determined by Longo et al. (2020) for lakes in Northern Alaska where the lake model was validated using limnological data from the Toolik Environmental Data Center and the Arctic Long-Term Ecological Research program over a 6-year period (see Table A1). Lake E5 in Northern Alaska is similar in size and with a similar catchment area to VGHV (Longo et al., 2020). The neutral drag coefficient was set to 0.002, and the albedos for slush and snow were set to 0.4 and 0.7, respectively. Note that volcanic eruptions in Iceland can result in ash deposits on the snow, and lower the albedo of the resulting slush and snow cover on VGHV and lead to earlier ice-off dates (Landl et al., 2003). However, we expect this to only be important during volcanic eruptions that occurred during the winter and/or spring season and would only influence the lake water temperatures and ice-cover during that year. As our purpose is to understand how the lake responds on longer-timescales, we keep the values for albedo constant. The model was initialized using ERA-Interim daily data (1979-2018 CE; ECMWF; Dee et al., 2011) averaged over grid cells covering southwest Iceland (18.25° W-22.75° W by 63.00° N-64.50° N for a 0.75° x 0.75° grid). An initial control simulation was run for 39 years, followed by sensitivity tests where various perturbations were introduced. Results from the control simulation were compared with available meteorological data, ice-off dates from nearby lakes in Iceland, and the few observations we were able to make using satellite imagery when our study site was not obscured by clouds (Tables A2 & A3; Fig. A2). The lack of extensive observational data from VGHV prevents us from validating the outputs from the lake model simulation, therefore we use the outputs from the lake model to highlight

what processes could lead to variations in ice-off dates and lake water temperatures during the spring season, but not to quantify the number of days or degrees that ice-off dates and temperatures, respectively, changed in the lake over the last 2,000 years." (section 2.4, lines 166-187)

"Currently there are no extensive datasets on changes in lake water temperatures and/or ice-off dates for lakes (in particular lakes that are not influenced by geothermal activity or are glacial lakes that are subject to sea water intrusions) in Iceland to validate the control simulation from our lake model. However, a comparison with existing data in the literature and satellite images that were not obscured by cloud-cover suggests that the timing of the ice-out dates in our lake model (mid- to late April) and mean monthly temperatures of the surface lake water are reasonable (Tables A2 & A3; Fig. A2). We use the results from the lake model to infer what processes could drive large changes in ice-off dates and lake water temperatures, and thereby the determine the seasonal sensitivity of our proxy." (section 3.2, lines 240-246)

***Reviewer:*** Group I studies: both me, but especially the other reviewer, ask for a bit more regional context about the regional application of the UK37 proxy. But instead of adding certain relevant recent studies, a number of papers have been left out. As some of these are mentioned in the rebuttal, I suspect something as gone awry here. I urge you to have a look at this: the most striking examples (previously mentioned, but now somehow missing) include 1) Harning et al. 2020 GRL, and 2) van der Bilt et al. 2020 GRL.

***Response:*** We added additional text to section 2.3 to further discuss the application and seasonality of the $U_{37}^K$ index in lakes:

"Temperature calibrations using the $U_{37}^K$ index were applied to develop high resolution records of summer temperatures in Greenland (c. 5,600 yrs BP; D'Andrea et al., 2011) and Svalbard (1,800 yrs BP; D'Andrea et al., 2012 and 12,000 yrs BP; van der Bilt et al., 2018, 2019), as well as a winter-spring temperature record in Alaska (16,000 yrs BP; Longo et al., 2020). The main timing of ice-off, and the corresponding alkenone bloom in VGHV, occurs in April to May in southern Iceland and May to June in northern Iceland (see Tables A2 & A3). In contrast, studies in Svalbard report ice-off dates between late June to mid-August and show that alkenones primarily record summer (JJA) temperatures (D'Andrea et al., 2012; van der Bilt et al., 2019). In Northern Alaska ice-off dates primarily occur in June and reflect temperature changes during the winter and spring season (Longo et al., 2018, 2020). The regional variability in the relationship between the $U_{37}^K$ index and temperature, as well as differences in the timing of the alkenone bloom requires the development of local temperature calibrations and validation of the seasonal sensitivity of the proxy for each region (Wang and Liu, 2013; D'Andrea et al., 2016; Longo et al., 2016). Unfortunately, there is currently no local calibration for Icelandic lakes, but in the following section we will describe how we use a lake model to test what drives changes in ice-off dates and lake water temperatures, and therefore the seasonality of temperature recorded by alkenones, in VGHV." (section 2.3, lines 146-158)

We would also like to explain why we chose not to elaborate on the study by Harning et al. (2020 GRL) in our discussion. The study by Harning et al. (2020) does not include an in-situ calibration to determine the seasonality of alkenones at the study site, Lake Skorarvatn, in Iceland, although it was mentioned as future work. The seasonality of alkenones is assumed to be summer based on the seasonality of alkenones produced in

Svalbard and Greenland (ice-out in late June to August; D'Andrea et al., 2011, 2012), even though ice-off at Lake Skorarvatn occurs in mid-April to mid-May (Harning et al., 2020). In Northern Alaska, where ice-off occurs in June, alkenones were rigorously shown to record winter-spring temperatures (Longo et al., 2018, 2020). This highlights the need for a more rigorous validation of how spring lake water temperatures and the timing of ice-off in Skorarvatn, and therefore the temperatures recorded by alkenones, respond to seasonal changes in air temperature. The calibration used for reconstructing temperatures downcore relies on the slope for Group I Isochrysidales determined by D'Andrea et al. (2016) and is used to calculate temperature anomalies over the Holocene. The record consists of 10 datapoints for alkenones over the last 10,000 years and only two data points over the last 2,000 years. Unfortunately, this makes it difficult to draw any concrete conclusions on the seasonality of the proxy or what is driving long-term changes in Icelandic temperatures. Hopefully, future studies will be able to expand on both our study and the work by Harning et al. (2020) to develop new calibrations for alkenones in Iceland and better constrain the production and seasonality of this proxy.

*Reviewer:* Calibration: you argue that most alkenones in your site derive from Group I producers, remove samples that (also) contain Group II, and yet opt for a Group III temperature calibration because these values are "more reasonable". I don`t find this a particularly strong line of argumentation, and think you need to discuss this decision in more detail. Key element: available Group I calibrations now derive from many NH lakes and produce calibration slopes that are rather similar, so why would VGHV be so different?

*Response:* We thank the reviewer for pointing this out, and we apologize for not clarifying this statement. None of the existing calibrations generate reasonable temperatures at our study site. It is unclear why VGHV behaves differently. However, it is important to note that there are very few studies on Group I alkenones, and even in the Northern Hemisphere temperature calibration developed by Longo et al. (2018) there is considerable variation in the proxy (i.e., only 60% of the variability is explained by changes in spring temperatures). Further, differences in lake temperature sensitivity to air temperatures relative to previous studies could influence the resulting $U_{37}^{K}$ -temperature relationship. This is why we encourage the development of local alkenone calibrations for lakes in Iceland. We have modified the text in section 3.3 to make this clearer:

[revised manuscript text omitted]

**Reviewer 2**

*Reviewer:* Richter et al. present a revised and nicely improved manuscript, with the major accomplishments of the manuscript in place from the initial submission and many of my primary concerns from the original manuscript addressed. In particular, I find the modified discussion around the seasonality of the alkenone proxy to be very clear and convincing, given the available calibration data.

*Response:* Thank you for taking the time to review our manuscript and for your feedback

*Reviewer:* While I would suggest some further revision for clarity, my criticisms of the current manuscript are mostly minor and I recommend the manuscript for publication:

One of the primary conclusions reads as that increasing winter-spring insolation over the common era drove significant cold-spring warming at VGHV. However, the lake model they generate shows very little ice-off date/water temperature response to the amount of winter-spring insolation change over the common era. While this model does show a strong response to temperature, they haven't really explained how that change in insolation could drive that kind of temperature change (do they attribute the trend then to local feedbacks? Other climate process? driven by insolation). In my opinion, it would improve the manuscript if the authors are able to explain this aspect of their interpretation in very clear way. On this point, I find the discussion around drivers of climate around Iceland and the NH (section 4) somewhat meandering. It was a little bit of a challenge for me to tease out the important take-aways, and I think the section could be improved with a little bit of restructuring for clarity and flow.

*Response:* Thank you for pointing this out. To improve the clarity of our discussion, we decided to split section 4.1 into two different discussion sections. The first section (now 4.1) focuses on regional drivers of seasonal climate change in Iceland, and in particular how this impacts our study site and what implications this has for the interpretation of our proxy record. The next section (now 4.2) aims to put our record into the context of broader changes in seasonal climate changes observed in the NH both during the Holocene and the last 2,000 years.

*Reviewer:* -Given the qualitative nature of the proxy, I am somewhat unconvinced that relatively small trends outside of the long-term warming trend are climatically meaningful. The authors dedicate a good amount of text to discussing multi-decadal to centennial scale trends (section 4.2) but do not put these into the context of error on the proxy (are these changes larger than the error on any of the calibration data? Even if they are, can we trust that error range in a region that has no local calibration data?). It strikes me as an unnecessary and under supported component of the paper. It seems to me like it would be a better use of available space to clearly explain why, if the primary driver is insolation, from 0-400 CE does not seem to actually track with winter-spring insolation.

*Response:* We thank the reviewer for bringing this to our attention. Although we are unable to quantify the amplitude of temperature change in our record, the $U^K_{37}$ index is known to be sensitive to changes in temperature in NH lakes on millennial and multi-decadal timescales (see sections 2.3 and 3.3). The variability observed in the $U^K_{37}$ index in our record falls well outside the analytical uncertainty of the alkenone standard that was measured throughout our analyses ($\sigma = 0.0040$, n=28). Therefore, after correcting for species-mixing affects, we

assume that temperature is the only environmental factor that will lead to significant deviations in the $U^K_{37}$ index. We added the following sentences to section 2.2:

"An alkenone standard was injected twice every 8 to 10 samples to assess the analytical precision of the alkenone measurements. The standard deviation for the calculated $U^K_{37}$ index (see equation 1) was 0.0040 (n = 28)." (section 2.2, lines 100-102)

We modified the text in section 3.3 as follows:

"Despite these differences, the $U^K_{37}$ index is known to be highly sensitive to temperatures in NH lakes and both the millennial and multidecadal variability in our $U^K_{37}$ index exceeds the range of our analytical uncertainty. We assume that, after correcting for species mixing, there are not environmental parameters other than temperature that affect our $U^K_{37}$ record, and we therefore use the $U^K_{37}$ index to infer and evaluate qualitative changes in temperature trends and variability during the past 2,000 years." (section 3.3, lines 281-285)

We use section 4.3 (formerly section 4.2) to discuss the variability of the $U^K_{37}$ index in the context of forcings that are important in the North Atlantic region and to compare our record with existing summer and mean annual paleoclimate records. We have shortened section 4.3 and modified the text to focus on larger climate anomalies during the last 2,000 years and discuss how these are manifested in our record and offer a possible explanation as to why we observe cooling or warming during these time intervals. To address your point concerning the observed cooling between 0-400 CE followed by warming in our record, we have added the following sentences to section 4.3 as a possible explanation:

"The heterogenous temperature response in the North Atlantic region could be associated with a strengthening of the SPG between c. 200-400 CE, resulting in increased oceanic meridional heat transport to Northern Europe and colder SSTs near southern Iceland (Miettinen et al., 2012; Moffa-Sánchez and Hall, 2017). After c. 400 CE there is evidence of increased salinity and warming SSTs south of Iceland, that are associated with a gradual weakening of the SPG and contributed to a return to warmer temperatures in our VGHV record (Thornalley et al., 2009; Moffa-Sánchez and Hall, 2017; Moreno-Chamarro et al., 2017). A weaker SPG after c. 400 CE would also result in cooler Nordic waters and contribute to the cooler climate conditions observed in Northern Europe c. 500-650 CE (Miettinen et al., 2012; Helama et al., 2017; Moffa-Sánchez and Hall, 2017)." (section 4.3, lines 420-427)

*Reviewer:* And finally, a note to the authors to check the caption on Fig. 5 prior to final publication.

*Response:* Thank you for pointing this out. We fixed the caption in Fig. 5.